# INFORMATION-THEORETIC ACTIVE CORRELATION CLUSTERING

## ABSTRACT

We study correlation clustering where the pairwise similarities are not known in advance. For this purpose, we employ active learning to query pairwise similarities in a cost-efficient way. We propose a number of effective information-theoretic acquisition functions based on entropy and information gain. We extensively investigate the performance of our methods in different settings and demonstrate their superior performance compared to the alternatives.

## 1 INTRODUCTION

Clustering is an important unsupervised learning problem for which several methods have been proposed in different contexts. *Correlation clustering* (CC) (Bansal et al., 2004; Demaine et al., 2006) is a well-known clustering problem, especially beneficial when both similarity and dissimilarity assessments exist for a given set of $N$ objects. Consequently, CC studies the clustering of objects where pairwise similarities can manifest as positive or negative numbers. It has found a wide range of applications including image segmentation (Kim et al., 2011), bioinformatics (Bonchi et al., 2013), spam filtering (Ramachandran et al., 2007; Bonchi et al., 2014), social network analysis (Bonchi et al., 2012; Tang et al., 2016), duplicate detection (Hassanzadeh et al., 2009), co-reference identification (McCallum & Wellner, 2004), entity resolution (Getoor & Machanavajjhala, 2012), color naming across languages (Thiel et al., 2019) and clustering aggregation (Gionis et al., 2007; Chehreghani & Chehreghani, 2020). CC was initially explored using binary pairwise similarities in $\{-1, +1\}$ (Bansal et al., 2004), and was later extended to support arbitrary positive and negative pairwise similarities in $\mathbb{R}$ (Charikar et al., 2005; Demaine et al., 2006). Finding the optimal solution for CC is known to be NP-hard and APX-hard (Bansal et al., 2004; Demaine et al., 2006), presenting significant challenges. As a result, various approximate algorithms have been developed to address this problem (Bansal et al., 2004; Charikar et al., 2005; Demaine et al., 2006; Ailon et al., 2008; Elsner & Schudy, 2009). Among these, methods based on local search are noted for their superior performance in terms of clustering quality and computational efficiency (Thiel et al., 2019; Chehreghani, 2023).

Existing methods generally assume that all $\binom{N}{2}$ pairwise similarities are available beforehand. However, as discussed in (Bressan et al., 2019; García-Soriano et al., 2020), generating pairwise similarities can be computationally intensive and may need to be obtained through resource-intensive queries, e.g., from a human expert. For instance, determining interactions between biological entities often requires the expertise of highly trained professionals, consuming both time and valuable resources (García-Soriano et al., 2020). In tasks like entity resolution, obtaining pairwise similarity queries through crowd-sourcing could also involve monetary costs. Therefore, a central question emerges: *How can we design a machine learning paradigm that effectively delivers satisfactory CC results with a limited number of queries for pairwise similarities between objects?*

In machine learning, *active learning* is generally employed to address such a question. Its objective is to acquire the most informative data within a constrained budget. Active learning has proven effective in various tasks, including recommender systems (Rubens et al., 2015), sound event detection (Shuyang et al., 2020), analysis of driving time series (Jarl et al., 2022), drug discovery (Viet Johansson et al., 2022), and analysis of logged data (Yan et al., 2018). In the context of active learning, the selection of which data to query is guided by an *acquisition function*. Active learning is most commonly studied for classification and regression problems (Settles, 2009). However, it has also been studied for clustering and is sometimes referred to as *supervised clustering* (Awasthi & Zadeh, 2010). The objective is to discover the ground-truth clustering with a minimal number of

queries to an *oracle* (e.g., a human expert). In this scenario, queries are typically executed in one of two ways: (i) By asking whether two clusters should merge or if one cluster should be divided into multiple clusters (Balcan & Blum, 2008; Awasthi & Zadeh, 2010; Awasthi et al., 2017); (ii) By querying the pairwise relations between objects (Basu et al., 2004; Mazumdar & Saha, 2017b;a; Saha & Subramanian, 2019; Bressan et al., 2019; García-Soriano et al., 2020; van Craenendonck et al., 2018b; Silwal et al., 2023; Gullo et al., 2023; Aronsson & Chehreghani, 2024; Kuroki et al., 2024).

Among the aforementioned works on active learning for clustering, only (Mazumdar & Saha, 2017b; Bressan et al., 2019; García-Soriano et al., 2020; Aronsson & Chehreghani, 2024; Kuroki et al., 2024) consider the setting that we are interested in: (i) The clustering algorithm is based on CC; (ii) The pairwise similarities are not assumed to be known in advance; (iii) We assume access to a single noisy oracle, to which a *fixed* budget $B \ll \binom{N}{2}$ of queries for pairwise similarities can be performed; (iv) Access to feature vectors is not assumed by the algorithm, meaning that information about the ground-truth clustering is solely obtained through querying the oracle for pairwise similarities. Throughout the paper, this setting will be referred to as *active correlation clustering*.

The work in (Mazumdar & Saha, 2017b) develops a number of *pivot-based* CC algorithms that satisfy guarantees on the query complexity, assuming a noisy oracle. However, the algorithms are purely theoretical and are not implemented and investigated in practice, and require setting a number of non-trivial parameters (e.g., they assume the noise level is known in advance which is unrealistic). The work in (Bressan et al., 2019; García-Soriano et al., 2020) proposes adaptive and query-efficient versions of the simple pivot-based CC algorithm KwikCluster (Ailon et al., 2008). However, as demonstrated in (Aronsson & Chehreghani, 2024), such pivot-based methods perform very poorly for active CC with noise. The work in (Gullo et al., 2023; Kuroki et al., 2024) address query-efficient CC by formulating it as a *multi-armed bandit* problem. However, this leads to a number of limiting assumptions in practice. We defer a detailed comparison to Appendix E.

The work in (Aronsson & Chehreghani, 2024) proposes a generic active CC framework that overcomes the limitations of previous work and offers several advantages: (i) The pairwise similarities can be any positive or negative real number, even allowing for inconsistencies (i.e., violation of transitivity). This allows the oracle to express uncertainty in their feedback; (ii) The process of querying pairwise similarities is decoupled from the clustering algorithm, enhancing flexibility in constructing acquisition functions that can be employed in conjunction with *any* CC algorithm. (Aronsson & Chehreghani, 2024) employs an efficient CC algorithm based on local search, whose effectiveness (and superiority over pivot-based methods) has also been demonstrated in the standard CC setting (Thiel et al., 2019; Chehreghani, 2023), and dynamically computes the number of clusters; (iii) The framework is robust w.r.t. a noisy oracle and supports multiple queries for the same pairwise similarity if needed (to deal with noise).

Furthermore, (Aronsson & Chehreghani, 2024) proposes two novel acquisition functions, namely *maxmin* and *maxexp*, to be used within their framework. They demonstrate that the algorithm QECC from (García-Soriano et al., 2020) performs poorly in the presence of even a very small amount of noise and is significantly outperformed by their methods. In this paper, we adopt the generic active CC framework in (Aronsson & Chehreghani, 2024) with a focus on the development of more effective acquisition functions. The contributions of this paper are the following:

- We investigate the use of information-theoretic acquisition functions based on *entropy* and *information gain* for active CC. We propose four different acquisition functions inspired by this (see Section 3). Although information-theoretic acquisition functions have been extensively studied in the context of active learning (Roy & McCallum, 2001; Kirsch & Gal, 2022), prior research has focused mainly on (active) *supervised learning* scenarios, where the goal is to query data labels from an oracle rather than pairwise relations. *To our knowledge, our work is the first attempt to propose information-theoretic acquisition functions to active learning with pairwise relations, as well as to non-parametric models like CC.* Computing the necessary quantities in this setting is significantly more complex. The methods proposed in this paper can be applied beyond active CC, including to the active learning of other pairwise (non-parametric) clustering models.

- We conduct extensive experimental studies on various datasets that demonstrate the superior performance of our acquisition functions compared to *maxmin* and *maxexp* (and other baselines), and investigate a number of interesting insights about the active CC framework from (Aronsson & Chehreghani, 2024) (see Section 4 and Appendix C).

## 2 ACTIVE CORRELATION CLUSTERING

In this section, we begin by introducing the problem of active CC. After this, we describe the active clustering procedure used to solve this problem.

### 2.1 PROBLEM FORMULATION

We are given a set of $N$ objects (data points) indexed by $\mathcal{V} = \{1, \ldots, N\}$. The set of pairs of objects in $\mathcal{V}$ is denoted by $\mathcal{E} = \{(u, v) \mid u, v \in \mathcal{V}\}$. We assume the existence of a ground-truth similarity matrix $\boldsymbol{S}^* \in \mathbb{R}^{N \times N}$, which represents the true pairwise similarities between every pair $(u, v) \in \mathcal{E}$. However, $\boldsymbol{S}^*$ is not known beforehand. Instead, one can only query the oracle for a noisy version of this matrix for a desired pair of objects, while incurring some cost. We use $\boldsymbol{S} \in \mathbb{R}^{N \times N}$ to represent an estimate of the pairwise similarities. If $S_{uv} = S_{uv}^*$ for all $(u, v) \in \mathcal{E}$ we have a perfect estimate of the true pairwise similarities, which we assume is unrealistic in practice. Hence, the objective is to discover the ground-truth clustering solution with a minimal number of (active) queries for the pairwise similarities to the oracle, since each query incurs some cost. A similarity matrix $\boldsymbol{S}$ is symmetric, and we assume zeros on the diagonal, i.e., $S_{uv} = S_{vu}$ and $S_{uu} = 0$. This means there are $\binom{N}{2} = (N \times (N-1))/2$ unique pairwise similarities to estimate. Without loss of generality, we assume all similarities are in the range $[-1, +1]$. In this case, $+1$ and $-1$ respectively indicate definite similarity and dissimilarity. Thus, a similarity close to $0$ indicates a lack of knowledge about the relation between the two objects. This allows the oracle to express uncertainty in their feedback.

A clustering is a partition of $\mathcal{V}$. In this paper, we encode a clustering with $K$ clusters as a clustering solution $\boldsymbol{c} \in \mathbb{K}^N$ where $\mathbb{K} = \{1, \ldots, K\}$ and $c_u \in \mathbb{K}$ denotes the cluster label of object $u \in \mathcal{V}$. We denote by $\mathcal{C}$ the set of clustering solutions for all possible partitions (clusterings) of $\mathcal{V}$. Given a clustering solution $\boldsymbol{c} \in \mathcal{C}$, the CC cost function $R^{\text{CC}} : \mathcal{C} \to \mathbb{R}^+$ aims to penalize cluster disagreements, as shown in Eq. 1.

$$R^{\text{CC}}(\boldsymbol{c} \mid \boldsymbol{S}) \triangleq \sum_{(u,v) \in \mathcal{E}} \begin{cases} |S_{uv}| & \text{if } (c_u = c_v \text{ and } S_{uv} < 0) \text{ or } (c_u \neq c_v \text{ and } S_{uv} \geq 0) \\ 0 & \text{otherwise.} \end{cases} \tag{1}$$

**Proposition 2.1.** *Eq. 1 can be simplified to $R^{CC}(\boldsymbol{c} \mid \boldsymbol{S}) = -\sum_{\substack{(u,v) \in \mathcal{E} \\ c_u = c_v}} S_{uv} + \text{constant}$, where the constant is independent of different clustering solutions (Chehreghani, 2013).*

All the proofs are in Appendix A. Based on Proposition 2.1, we define the *max correlation* cost function as

$$R^{\text{MC}}(\boldsymbol{c} \mid \boldsymbol{S}) \triangleq - \sum_{\substack{(u,v) \in \mathcal{E} \\ c_u = c_v}} S_{uv}, \tag{2}$$

and we have $\arg\min_{\boldsymbol{c} \in \mathcal{C}} R^{\text{CC}}(\boldsymbol{c} \mid \boldsymbol{S}) = \arg\min_{\boldsymbol{c} \in \mathcal{C}} R^{\text{MC}}(\boldsymbol{c} \mid \boldsymbol{S})$. Because of this, we will use $R^{\text{MC}}$ throughout most of the paper, as it leads to a number of simplifications in the presented methods. The conditioning on $\boldsymbol{S}$ for $R^{\text{CC}}$ and $R^{\text{MC}}$ will often be dropped, unless it is not clear from context. Finally, the ground-truth clustering solution corresponds to $\boldsymbol{c}^* = \arg\min_{\boldsymbol{c} \in \mathcal{C}} R^{\text{MC}}(\boldsymbol{c} \mid \boldsymbol{S}^*)$.

### 2.2 ACTIVE CORRELATION CLUSTERING PROCEDURE

We adopt the recent generic active CC procedure outlined in (Aronsson & Chehreghani, 2024) to solve the problem described in the previous section. The procedure is shown in Alg. 1. It takes an initial similarity matrix $\boldsymbol{S}^0$ as input, which can contain partial or no information about $\boldsymbol{S}^*$, depending on the initialization method. The procedure then follows a number of iterations, where each iteration $i$ consists of three steps: (i) *Update* the current clustering solution $\boldsymbol{c}^i \in \mathcal{C}$ by running a CC algorithm given the current similarity matrix $\boldsymbol{S}^i$. The current similarity matrix $\boldsymbol{S}^i$ will be referred to as $\boldsymbol{S}$ throughout the paper; (ii) *Select* a batch $\mathcal{B} \subseteq \mathcal{E}$ of pairs of size $B = |\mathcal{B}|$ based on an acquisition function $a : \mathcal{E} \to \mathbb{R}$. The quantity $a(u, v)$ indicates how informative the pair $(u, v) \in \mathcal{E}$ is, where a higher value implies greater informativeness. The optimal batch is selected by $\mathcal{B} = \arg\max_{\mathcal{B} \subseteq \mathcal{E}, |\mathcal{B}| = B} \sum_{(u,v) \in \mathcal{B}} a(u, v)$. This corresponds to selecting the top-$B$ pairs based on their acquisition value; (iii) *Query* the oracle for the pairwise similarities of the pairs $(u, v) \in \mathcal{B}$ and update each $S_{uv}^{i+1}$ based on the response.

---

**Algorithm 1** Active CC

---

1: **Input:** Initial similarity matrix $\boldsymbol{S}^0$, acquisition function $a$, batch size $B$.
2: $i \leftarrow 0$
3: **while** query budget not reached **do**
4:     $\boldsymbol{c}^i \leftarrow \text{CC}(\boldsymbol{S}^i)$                                                                ▷ Alg. 5
5:     $\mathcal{B} \leftarrow \arg\max_{\mathcal{B} \subseteq \mathcal{E}, |\mathcal{B}|=B} \sum_{(u,v) \in \mathcal{B}} a(u,v)$
6:     Query (noisy) oracle and update $S_{uv}^{i+1}$ for all pairs $(u,v) \in \mathcal{B}$
7:     $i \leftarrow i + 1$
8: **end while**
9: **return** $\boldsymbol{c}^i$

---

## 3   INFORMATION-THEORETIC ACQUISITION FUNCTIONS

In this section, we introduce four information-theoretic acquisition functions for active CC. All quantities defined below are conditioned on the current similarity matrix $\boldsymbol{S}$, but it is left out for brevity. All acquisition functions proposed in this section depend on the Gibbs distribution defined as

$$P^{\text{Gibbs}}(\mathbf{y} = \boldsymbol{c}) \triangleq \frac{\exp(-\beta R^{\text{MC}}(\boldsymbol{c}))}{\sum_{\boldsymbol{c}' \in \mathcal{C}} \exp(-\beta R^{\text{MC}}(\boldsymbol{c}'))}, \tag{3}$$

where $\beta \in \mathbb{R}^+$ is the concentration parameter, $\mathbf{y} = \{y_1, \ldots, y_N\}$ is a random vector with sample space $\mathcal{C}$ (all possible clustering solutions of $\mathcal{V}$) and $y_u$ is a random variable for the cluster label of $u$ with sample space $\mathbb{K}$. Computing $P^{\text{Gibbs}}$ is intractable due to the sum over all possible clustering solutions $\mathcal{C}$ in the denominator. Therefore, in the next section, we describe a *mean-field approximation* of $P^{\text{Gibbs}}$ which makes it possible to efficiently calculate the proposed acquisition functions. To the best of our knowledge, the use of mean-field approximation to approximate complex quantities when applying information-theoretic acquisition functions for active learning is a novel aspect of our approach. This approach can be applied beyond active CC, extending to the active learning of other pairwise, non-parametric clustering models.

### 3.1   MEAN-FIELD APPROXIMATION FOR CC

We here describe the mean-field approximation of $P^{\text{Gibbs}}$. The family of factorial distributions over the space of clustering solutions is defined as $\mathcal{Q} = \{Q \in \mathcal{P} \mid Q(\mathbf{y} = \boldsymbol{c}) = \prod_{u \in \mathcal{V}} Q(y_u = c_u)\}$, where $\mathcal{P}$ is the space of all probability distributions with sample space $\mathcal{C}$. The goal of mean-field approximation is to find a factorial distribution $Q \in \mathcal{Q}$ that best approximates the intractable distribution $P^{\text{Gibbs}}$. In general, one can compute the optimal $Q$ by minimizing the KL-divergence (Hofmann & Buhmann, 1997; Chehreghani et al., 2012), i.e.,

$$Q^* = \underset{Q \in \mathcal{Q}}{\arg\min} \, D_{\text{KL}}(Q \| P^{\text{Gibbs}}) = \underset{Q \in \mathcal{Q}}{\arg\min} \sum_{\boldsymbol{c} \in \mathcal{C}} Q(\boldsymbol{c}) \log \frac{Q(\boldsymbol{c})}{P^{\text{Gibbs}}(\boldsymbol{c})}. \tag{4}$$

We encode a mean-field approximation using a matrix of assignment probabilities $\boldsymbol{Q} \in [0,1]^{N \times K}$, where $Q_{uk} = Q(y_u = k)$. In addition, let $\boldsymbol{M} \in \mathbb{R}^{N \times K}$, where $M_{uk}$ should be interpreted as the cost of assigning object $u$ to cluster $k$. Given this, Theorem 3.1 implies that an EM-type procedure, which sequentially alternates between estimating $Q_{uk}$ (based on Eq. 5) and computing the respective $M_{uk}$ (based on Eq. 6), yields a local minimum for the optimization problem in Eq. 4. In Theorem 3.1, we adapt and specialize the general result from (Hofmann et al., 1998) to our specific cost function in Eq. 2, enabling efficient mean-field approximations tailored to our model, which are essential for all proposed acquisition functions.

**Theorem 3.1.** *Let $\ell : \mathbb{N} \to \mathcal{V}$ denote an object visitation schedule, which satisfies $\lim_{T \to \infty} |\{t \le T : \ell(t) = u\}| = \infty, \forall u \in \mathcal{V}$. For arbitrary initial conditions, the asynchronous update rules defined by*

$$Q_{uk}^{(t+1)} = \exp(-\beta M_{uk}^{(t)}) / \sum_{k' \in \mathbb{K}} \exp(-\beta M_{uk'}^{(t)}), \tag{5}$$

$$M_{uk}^{(t+1)} = -\sum_{\substack{v \in \mathcal{V} \\ v \neq u}} S_{uv} Q_{vk}^{(t+1)}, \tag{6}$$

*where $u = \ell(t)$, converge to a local minimum of Eq. 4.*

For computational efficiency, we employ a synchronous update rule in practice (see Alg. 2). Despite not having the same theoretical guarantees, synchronous updates have been observed to perform well empirically in other contexts (Hofmann et al., 1998; Chehreghani et al., 2012). Alg. 2 assumes a fixed number of clusters $K$. We use the number of clusters $K$ dynamically determined by the CC algorithm used at each iteration $i$ of Alg. 1 to find $c^i$ (see Appendix D for details of this algorithm). $M$ could be

---

**Algorithm 2** Mean-Field Approximation

1: **Input:** Similarity matrix $S$, cluster assignment costs $M$, concentration parameter $\beta$.
2: **while** $Q$ has not converged **do**
3:     $Q \leftarrow \text{softmax}(-\beta M)$         ▷ E-step
4:     $M \leftarrow -S \cdot Q$         ▷ M-step
5: **end while**
6: **return** $Q, M$

---

initialized randomly. However, since we have the current clustering solution $c^i$, we initialize it based on $c^i$, i.e., $M_{uk} = -\sum_{v:c_v^i=k} S_{uv}$, in order to speed up the convergence and potentially improve the quality of the solution found. This initialization of $M$ is based on the total similarity between object $u$ and cluster $k$ in relation to the similarity between $u$ and all other clusters. A smaller similarity should correspond to a higher cost (hence the negation). Each iteration of the algorithm consists of two main steps. First, $Q$ is estimated as a function of $M$. Second, $M$ is calculated based on $Q$. In this paper, we treat the concentration parameter $\beta \in \mathbb{R}^+$ as a hyperparameter. Finally, we employ the special form of the max correlation cost function $R^{\text{MC}}$ in Eq. 2, and calculate both the E-step and M-step in vectorized form. In particular, the M-step becomes a dot product between $S$ and $Q$, which is extremely efficient in practice (especially if $S$ is assumed sparse, which it is in our experiments).

### 3.2 Entropy

In this section, we propose our first acquisition function based on entropy. Let $\mathbf{E} \in \{-1, +1\}^{N \times N}$ be a random matrix where each element $\mathrm{E}_{uv} \in \{-1, +1\}$ is a binary random variable, where $+1$ indicates $u$ and $v$ should be in the same cluster, and $-1$ implies $u$ and $v$ should be in different clusters. A reasonable way to define the probability of $\mathrm{E}_{uv}$ to be $+1$ is the fraction of clustering solutions in $\mathcal{C}$ that assign $u$ and $v$ to the same cluster, weighted by the probability of each clustering solution. Due to the intractability of $P^{\text{Gibbs}}$, we approximate it using a mean-field approximation $Q$ (encoded by matrix $Q$, as described in the previous section). Formally, we have

$$P(E_{uv} = 1) = \mathbb{E}_{P^{\text{Gibbs}}(\mathbf{y})}[\mathbf{1}_{\{y_u=y_v\}}] \approx \mathbb{E}_{Q(\mathbf{y})}[\mathbf{1}_{\{y_u=y_v\}}] = \sum_{k \in \mathbb{K}} Q_{uk} Q_{vk}. \tag{7}$$

The last equality of Eq. 7 uses the fact that the mean-field approximation assumes independence between objects. One can similarly derive $P(\mathrm{E}_{uv} = -1) = \sum_{k,k' \in \mathbb{K}} Q_{uk} Q_{vk'} \mathbf{1}_{\{k \neq k'\}} = 1 - P(\mathrm{E}_{uv} = 1)$. For more information on the calculations, see Appendix B.1. Thereby, from Eq. 7, we define an acquisition function based on the entropy of $\mathrm{E}_{uv}$ as

$$a^{\text{Entropy}}(u, v) \triangleq H(\mathrm{E}_{uv}) = \mathbb{E}_{P(\mathrm{E}_{uv})}[-\log P(\mathrm{E}_{uv})]. \tag{8}$$

### 3.3 Information Gain

The acquisition function $a^{\text{Entropy}}$ calculates the uncertainty of pairs based on the mean-field approximation (model) $Q$ given the current similarity matrix $S$. In this section, we investigate acquisition functions inspired by the notion of *information gain* corresponding to maximal uncertainty reduction. In this case, the similarity matrix $S$ is first augmented with *pseudo-similarities* (predicted using the current model $Q$ as $S_{uv} \sim P(\mathrm{E}_{uv})$), after which a new mean-field approximation is obtained. In other words, we simulate the effect of querying one or more pairs in expectation w.r.t. the current model $Q$, potentially resulting in more accurate uncertainty estimations. Due to the efficiency of Alg. 2 (mean-field), one can afford to run it several times per iteration of the active CC procedure, to estimate the information gain accurately. In this paper, we consider two types of information gain. First, the information gain (or equivalently the mutual information) between a pair $\mathrm{E}_{uv}$ and the cluster labels of objects $\mathbf{y}$. Due to symmetry of the mutual information we have

$$I(\mathbf{y}; \mathrm{E}_{uv}) = H(\mathbf{y}) - H(\mathbf{y} \mid \mathrm{E}_{uv}) \tag{9}$$
$$= H(\mathrm{E}_{uv}) - H(\mathrm{E}_{uv} \mid \mathbf{y}). \tag{10}$$

The interpretation of $I(\mathbf{y}; \mathrm{E}_{uv})$ is the amount of information one expects to gain about the cluster labels of objects by observing $\mathrm{E}_{uv}$, where the expectation is w.r.t. $P(\mathrm{E}_{uv})$. In other words, it measures

the expected reduction in uncertainty (in entropic way) over the possible clustering solutions w.r.t. the value of $\mathrm{E}_{uv}$. Second, the information gain between a pair $\mathrm{E}_{uv}$ and all pairs $\mathbf{E}$:

$$I(\mathbf{E}; \mathrm{E}_{uv}) = H(\mathbf{E}) - H(\mathbf{E} \mid \mathrm{E}_{uv}) \tag{11}$$

$$= H(\mathrm{E}_{uv}) - H(\mathrm{E}_{uv} \mid \mathbf{E}). \tag{12}$$

Intuitively, $I(\mathbf{E}; \mathrm{E}_{uv})$ measures the amount of information the pair $\mathrm{E}_{uv}$ provides about all pairs in $\mathbf{E}$. All the expressions above are closely related, but the formulations used will impact how they can be approximated in practice, leading to differences in performance and efficiency. This is discussed in detail in the following subsections.

### 3.3.1 CONDITIONAL MEAN-FIELD APPROXIMATION

We approximate all the conditional entropies defined above following the same general principle: We update the similarity matrix $\boldsymbol{S}$ based on what is being conditioned on, run Alg. 2 given this similarity matrix, and calculate the corresponding entropy given the updated mean-field approximation. Motivated by this, the following notation will be used throughout this section. Let $\boldsymbol{e}$ denote a vector in $\{-1, +1\}^{|\mathcal{D}|}$, where $\mathcal{D} \subseteq \mathcal{E}$ is a subset of the pairs. Given this, we denote by $\boldsymbol{Q}^{(\boldsymbol{S}_{\mathcal{D}}=\boldsymbol{e})}$ to be the mean-field approximation found by Alg. 2 after modifying $\boldsymbol{S}$ according to $\boldsymbol{e}$ for all pairs $(u, v) \in \mathcal{D}$ (with remaining pairs unchanged).

### 3.3.2 EXPECTED INFORMATION GAIN

In this section, we consider the expression in Eq. 9, which corresponds to the *expected information gain over cluster labels of objects* (EIG-O). We have $H(\mathbf{y} \mid \mathrm{E}_{uv}) = \mathbb{E}_{e \sim P(\mathrm{E}_{uv})}[H(\mathbf{y} \mid \mathrm{E}_{uv} = e)]$. In this paper, we approximate $H(\mathbf{y} \mid \mathrm{E}_{uv} = e)$ using conditional mean-field approximation $\boldsymbol{Q}^{(S_{uv}=e)}$ as shown below. Given some mean-field approximation $\boldsymbol{Q}'$, let $P(\mathrm{E}_{uv} \mid \boldsymbol{Q}')$ be the probability of $\mathrm{E}_{uv}$ computed as shown in Eq. 7 using $\boldsymbol{Q}'$ and

$$H(\mathrm{y}_w \mid \boldsymbol{Q}') \triangleq - \sum_{k \in \mathbb{K}} Q'_{wk} \log Q'_{wk}.$$

Each $\mathrm{y}_u \in \mathbf{y}$ is independent of other cluster labels given a mean-field approximation, reducing all joint entropies to the summed entropy across all individual variables. In other words, we have $H(\mathbf{y}) \approx \sum_{w \in \mathcal{V}} H(\mathrm{y}_w \mid \boldsymbol{Q})$ and $H(\mathbf{y} \mid \mathrm{E}_{uv} = e) \approx \sum_{w \in \mathcal{V}} H(\mathrm{y}_w \mid \boldsymbol{Q}^{(S_{uv}=e)})$. Given this, we define the following acquisition function.

$$a^{\text{EIG-O}}(u, v) \triangleq \sum_{w \in \mathcal{V}} H(\mathrm{y}_w \mid \boldsymbol{Q}) - \sum_{e \in \{-1, +1\}} P(\mathrm{E}_{uv} = e \mid \boldsymbol{Q}) H(\mathrm{y}_w \mid \boldsymbol{Q}^{(S_{uv}=e)}). \tag{13}$$

In Appendix B.2, we include a detailed derivation of $a^{\text{EIG-O}}$. In addition, we derive an alternative acquisition function based on Eq. 11, which instead computes the expected reduction in entropy over pairs $\mathbf{E}$ (called $a^{\text{EIG-P}}$). However, this method expectedly performs similar to $a^{\text{EIG-O}}$ in practice[1], while being less efficient. Calculating $a^{\text{EIG-O}}$ for all pairs requires executing Alg. 2 $\binom{N}{2}$ times, which can be inefficient for large $N$. In Alg. 3, we illustrate how we compute $a^{\text{EIG-O}}$ in practice, improving its efficiency in the following ways. (i) We evaluate Eq. 13 only for a subset of the pairs $\mathcal{E}^{\text{EIG}} \subseteq \mathcal{E}$. We select this subset as the top-$|\mathcal{E}^{\text{EIG}}|$ pairs according to $a^{\text{Entropy}}$, where $|\mathcal{E}^{\text{EIG}}| = O(N)$ in practice. (ii) We do not expect $\boldsymbol{Q}^{(S_{uv}=e)}$ to be much different from $\boldsymbol{Q}$. Therefore, by initializing $\boldsymbol{M}$ (in lines 7-8) with the assignment costs from line 3, the convergence speed of Alg. 2 significantly improves.

### 3.3.3 JOINT EXPECTED INFORMATION GAIN

In this section, we consider the information gains formulated in Eq. 10 and Eq. 12. We approximate the conditional entropy $H(\mathrm{E}_{uv} \mid \mathbf{E}) = \mathbb{E}_{\boldsymbol{e} \sim P(\mathbf{E})}[H(\mathrm{E}_{uv} \mid \mathbf{E} = \boldsymbol{e})]$ in Eq. 12 using the conditional mean-field approximation $\boldsymbol{Q}^{(\boldsymbol{S}_{\mathcal{E}}=\boldsymbol{e})}$. The conditional entropy $H(\mathrm{E}_{uv} \mid \mathbf{y}) = \mathbb{E}_{\boldsymbol{c} \sim Q(\mathbf{y})}[H(\mathrm{E}_{uv} \mid \mathbf{y} = \boldsymbol{c})]$ in Eq. 10 is less straightforward here. However, a natural way would be to compute the conditional mean-field approximation given $\boldsymbol{S}$ updated based on $\boldsymbol{c}$ as follows. We set $S_{uv} = +1$ if $c_u = c_v$, and $-1$ otherwise. In both cases, we then approximate the entropy of $\mathrm{E}_{uv}$ given a mean-field

---

[1]This is expected because the distribution $P(\mathbf{E})$ is determined by $Q(\mathbf{y})$ from Eq. 7.

---

**Algorithm 3** EIG

---

1: **Input:** Similarity matrix $S$, current clustering $c^i$, concentration parameter $\beta$.
2: $M_{uk} \leftarrow -\sum_{v:c_v^i=k} S_{uv}, \forall u \in \mathcal{V}, \forall k \in \mathbb{K}$
3: $Q, M \leftarrow$ MeanField($S, M, \beta$)
4: $\mathcal{E}^{\text{EIG}} \leftarrow$ select top-$|\mathcal{E}^{\text{EIG}}|$ pairs using $a^{\text{Entropy}}$ given $Q$ (Eq. 8).
5: $a^{\text{EIG}}(u, v) \leftarrow 0, \forall (u, v) \in \mathcal{E}$
6: **for** each pair $(u, v) \in \mathcal{E}^{\text{EIG}}$ **do**
7: $\quad Q^{(S_{uv}=+1)} \leftarrow$ MeanField($S, M, \beta \mid S_{uv} = +1$)
8: $\quad Q^{(S_{uv}=-1)} \leftarrow$ MeanField($S, M, \beta \mid S_{uv} = -1$)
9: $\quad a^{\text{EIG}}(u, v) \leftarrow$ Evaluate Eq. 13 (or Eq. 34) given $Q$, $Q^{(S_{uv}=+1)}$ and $Q^{(S_{uv}=-1)}$
10: **end for**
11: **return** $a^{\text{EIG}}$

---

approximation conditioned on all (or a subset) of the similarities being updated. Given this, we now derive a general estimator based on the information gain

$$I(\mathrm{E}_{uv}; \mathbf{E}_{\mathcal{D}}) = H(\mathrm{E}_{uv}) - H(\mathrm{E}_{uv} \mid \mathbf{E}_{\mathcal{D}}), \tag{14}$$

where $\mathbf{E}_{\mathcal{D}} = \{\mathrm{E}_{uv} \mid (u, v) \in \mathcal{D}\}$ for some $\mathcal{D} \subseteq \mathcal{E}$. From the discussion above, the expressions in Eq. 10 and Eq. 12 can be seen as special cases of Eq. 14. The entropy $H(\mathrm{E}_{uv})$ is approximated based on Eq. 8. In addition, we have

$$\begin{aligned} H(\mathrm{E}_{uv} \mid \mathbf{E}_{\mathcal{D}}) &= \mathbb{E}_{e \sim P(\mathbf{E}_{\mathcal{D}})}[H(\mathrm{E}_{uv} \mid \mathbf{E}_{\mathcal{D}} = e)] \\ &= \sum_{e \in \{-1,+1\}^{|\mathcal{D}|}} P(\mathbf{E}_{\mathcal{D}} = e) H(\mathrm{E}_{uv} \mid \mathbf{E}_{\mathcal{D}} = e). \end{aligned} \tag{15}$$

Thereby, we approximate the conditional entropy $H(\mathrm{E}_{uv} \mid \mathbf{E}_{\mathcal{D}} = e)$ by

$$H(\mathrm{E}_{uv} \mid Q^{(S_{\mathcal{D}}=e)}) \triangleq - \sum_{e \in \{-1,+1\}} P(\mathrm{E}_{uv} = e \mid Q^{(S_{\mathcal{D}}=e)}) \log P(\mathrm{E}_{uv} = e \mid Q^{(S_{\mathcal{D}}=e)}). \tag{16}$$

In other words, we estimate the *joint* impact of pairs in $\mathcal{D}$ on the entropy of $\mathrm{E}_{uv}$. The expectation in Eq. 15, which involves a sum over all possible outcomes of $\mathbf{E}_{\mathcal{D}}$, quickly becomes intractable for large $|\mathcal{D}|$. However, one can easily obtain a sample $e \sim P(\mathbf{E}_{\mathcal{D}})$ by sampling $e_{uv} \sim P(\mathrm{E}_{uv})$ for every $\mathrm{E}_{uv} \in \mathbf{E}_{\mathcal{D}}$, which allows a Monte-Carlo estimation of this sum. For generality, assume we have $m$ subsets $\mathcal{D}_1, \ldots, \mathcal{D}_m, \mathcal{D}_i \subseteq \mathcal{E}$ and $n$ samples $e_i^1, \ldots, e_i^n \sim P(\mathbf{E}_{\mathcal{D}_i})$ for each $\mathcal{D}_i$. Given this, we then define the acquisition function

$$a^{\text{JEIG}}(u, v) \triangleq H(\mathrm{E}_{uv}) - \frac{1}{mn} \sum_{i=1}^{m} \sum_{l=1}^{n} H(\mathrm{E}_{uv} \mid Q^{(S_{\mathcal{D}_i}=e_i^l)}). \tag{17}$$

For $a^{\text{JEIG}}$, we only need to execute Alg. 2 (mean-field) $mn$ times, and in practice, we observe good performance with small values of $m$ and $n$. In Appendix B.3, we present Alg. 4 which describes the details of this method. Using $m$ subsets $\mathcal{D}_1, \ldots \mathcal{D}_m$ with each $|\mathcal{D}_i| \ll |\mathcal{E}|$ yields a number of benefits: (i) The Monte-Carlo estimation of the expectation in Eq. 15 becomes more accurate for smaller $n$ when $|\mathcal{D}_i|$ is smaller, which reduces the number of times Alg. 2 needs to be executed; (ii) If $\mathcal{D}_i = \mathcal{E}$, the conditional mean-field approximation $Q^{(S_{\mathcal{D}_i}=e)}$ is computed based on a similarity matrix where all pairs $(u, v) \in \mathcal{E}$ are sampled from $S_{uv} \sim P(\mathrm{E}_{uv})$, which will lead to extreme selection bias for the following reason: The probability $P(\mathrm{E}_{uv})$ (which is computed using $Q$) may already be biased (in particular in early iterations when $S$ contains incomplete/wrong information). Then, running Alg. 2 from scratch with a new similarity matrix fully augmented with biased information, will exaggerate the bias further; (iii) Using $m$ different subsets makes the estimator in Eq. 17 generic and flexible, but also captures more information about $\mathrm{E}_{uv}$, while remaining efficient and avoiding exaggerated selection bias.

## 4 EXPERIMENTS

In this section, we describe our experimental studies, where extensive additional results are presented in Appendix C.

## 4.1 EXPERIMENTAL SETUP

**Datasets.** In this paper, we use the datasets investigated by (Aronsson & Chehreghani, 2024): *20newsgroups*, *CIFAR10*, *cardiotocography*, *ecoli*, *forest type mapping*, *user knowledge modeling*, *MNIST* and *synthetic*. For all datasets, a random subset of at most $N = 1000$ objects are considered for the active CC experiments. See Appendix C.3 for details about all datasets.

**Correlation Clustering Algorithm.** We use the local search CC algorithm proposed by (Aronsson & Chehreghani, 2024) on line 4 of Alg. 1. It is highly robust to noise in $S$ and dynamically determines the number of clusters. The details of this algorithm are described in Appendix D.

**Ground-truth similarities.** For each experiment, we are given a dataset $\mathbf{X}$ with ground-truth labels $c^*$, where the ground-truth labels are only used for evaluations. Then, for each $(u, v) \in \mathcal{E}$ in a dataset, we set $S_{uv}^*$ to $+1$ if $u$ and $v$ belong to the same cluster, and $-1$ otherwise.

**Oracles.** We investigate four different oracles in Alg. 1: (i) **Oracle 1**. Returns $S_{uv}^*$ with probabiltiy $1 - \gamma$ or a uniform random value in $[-1, +1]$ with probability $\gamma$; (ii) **Oracle 2**. Returns a value sampled from $\mathcal{N}(S_{uv}^*, \gamma)$ (i.e., Gaussian centered at ground-truth similarity with variance $\gamma$); (iii) **Oracle 3**. Returns $S_{uv}^*$ with probabiltiy $1 - \gamma$ or $-S_{uv}^*$ with probability $\gamma$ (i.e., we flip the sign with probability $\gamma$)[2]; **Oracle 4**. We split the dataset into two disjoint parts $\mathbf{X} = \mathbf{X}_{\text{train}} \cup \mathbf{X}_{\text{test}}$. Then, we train a pairwise prediction model $f_\theta : \mathbf{X} \times \mathbf{X} \to [-1, +1]$ on $\mathbf{X}_{\text{train}}$, where ground-truth similarities $S^*$ are used as labels. Given any two data points $\mathbf{x}_u, \mathbf{x}_v \in \mathbf{X}$, we can predict their similarity as $f_\theta(\mathbf{x}_u, \mathbf{x}_v) \in [-1, +1]$. We then perform the CC experiments on data points in $\mathbf{X}_{\text{test}}$, and the oracle always returns the similarity $f_\theta(\mathbf{x}_u, \mathbf{x}_v)$. The ground-truth similarities of data points in $\mathbf{X}_{\text{test}}$ are never used when training $f_\theta$. We defer a detailed description of oracle 4 to Appendix C.2. The motivation for these oracles are as follows. Oracles 1-3 correspond to cases where the oracle provides unbiased information about $S^*$ (but noisy, with different noise models), allowing recovery of the ground-truth clustering solution $c^*$. This is considered by previous work (Mazumdar & Saha, 2017b; Silwal et al., 2023; Aronsson & Chehreghani, 2024). Oracle 4 may provide biased similarities due to noise/ambiguity in feature space, and exact recovery of $c^*$ may not be possible. This method is suggested by, e.g., (Bansal et al., 2004; Silwal et al., 2023) to compute pairwise similarities for CC.

**Initial similarities.** Let $\mathcal{E}^0$ be a uniform random subset of $\mathcal{E}$ (where $|\mathcal{E}^0| \ll |\mathcal{E}|$). Then, for all $(u, v) \in \mathcal{E}^0$, we initialize the current similarity matrix as $S_{uv}^0 = S_{uv}^*$ for oracles 1-3 and $S_{uv}^0 = f_\theta(\mathbf{x}_u, \mathbf{x}_v)$ for oracle 4. We then set $S_{uv}^0 = 0$ for $(u, v) \in \mathcal{E} \setminus \mathcal{E}^0$. Having $|\mathcal{E}^0| = 0$ or $|\mathcal{E}^0| > 0$ corresponds to a *cold-start* or *warm-start* setting, respectively. In this paper, like most previous work on active learning, we focus on a warm-start setting. See Appendix C.3 for the value of $|\mathcal{E}^0|$ for each dataset ($|\mathcal{E}^0|$ is chosen based on the size $N$ of each dataset).

**Repeated queries.** In general, Alg 1 supports multiple queries for the same pairwise similarity. This assumes each query for the same pair provides more information about the underlying distribution, which would be applicable to oracles 1-3. This is a common approach in active learning to deal with noisy oracles (Sheng et al., 2008; Settles, 2009). However, *we do not consider multiple queries for the same pair in our experiments*, as we found the difference in performance to be very small.

**Acquisition functions.** We have introduced four novel acquisition functions in this paper: $a^{\text{Entropy}}$ (Eq. 8), $a^{\text{EIG-O}}$ (Eq. 13), $a^{\text{EIG-P}}$ (Eq. 34) and $a^{\text{JEIG}}$ (Eq. 17). We compare these methods with *maxexp* and *maxmin* from (Aronsson & Chehreghani, 2024). In short, both *maxmin* and *maxexp* aim to query pairs with small absolute similarity that belong to triples $(u, v, w)$ that violate the transitive property of pairwise similarities. In other words, the goal is to reduce the inconsistency of $S$ by resolving violations of the transitive property in triples. In Appendix C.1, we include a detailed explanation of these methods. Finally, we include a simple baseline $a^{\text{Uniform}}(u, v) \sim \text{Uniform}(0, 1)$ which selects pairs uniformly at random. The work in (Aronsson & Chehreghani, 2024) compares maxexp/maxmin to a pivot-based active CC algorithm called QECC (García-Soriano et al., 2020) and two adapted state-of-the-art active constraint clustering methods, called COBRAS (van Craenendonck et al., 2018a) and nCOBRAS (Soenen et al., 2021). However, these methods perform very poorly compared to *maxexp* and *maxmin* in a noisy setting, so we exclude them here.

---

[2] Oracles 1-3 are equivalent if $\gamma = 0$. However, zero noise is unrealistic in practice. Also, it leads to fully consistent information in the similarities $S$, which makes the CC problem (minimization of Eq. 1) trivial (i.e., it is no longer NP-hard).

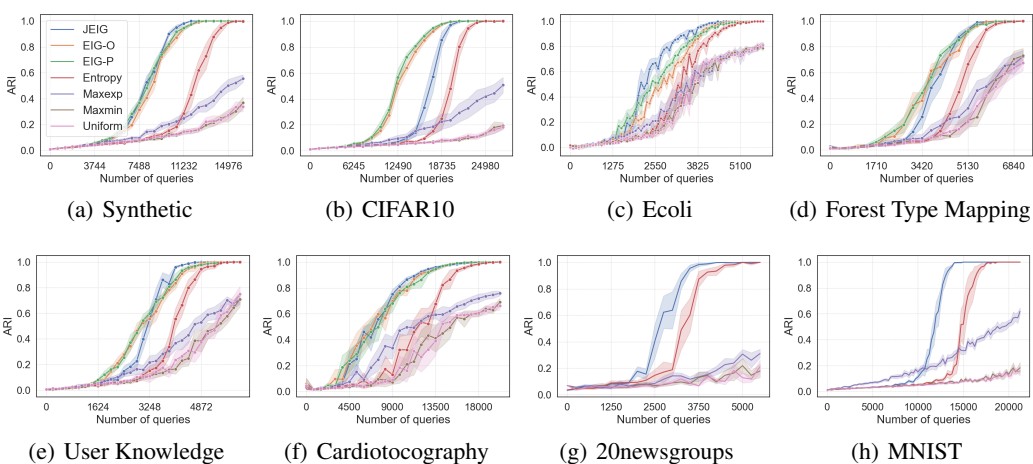

Figure 1: Results for oracle 1 with noise level $\gamma = 0.4$.

**Batch diversity.** In this paper, we consider single-sample acquisition functions that do not explicitly consider the joint informativeness among the elements in a batch $\mathcal{B}$. This has the benefit of avoiding the combinatorial complexity of selecting an optimal batch, which is a common problem for batch active learning (Ren et al., 2021). However, the work in (Kirsch et al., 2023) proposes a simple method for improving the batch diversity for single-sample acquisition functions using noise. In this paper, we utilize the *power* acquisition method. Given some acquisition function $a^{X}$, this corresponds to $a^{\text{PowerX}}(u, v) = \log(a^{X}(u, v)) + \epsilon_{uv}$ where $\epsilon_{uv} \sim \text{Gumbel}(0; 1)$. This is used for all information-theoretic acquisition functions proposed in this paper. We observe no benefit of this for maxmin/maxexp, likely due to their inherent randomness.

**Hyperparameters.** Unless otherwise specified, the following hyperparameters are used. The batch size $B$ depends on the dataset (since each dataset is of different size $N$). See Appendix C.3 for the value of $B$ for each dataset. For all information-theoretic acquisition functions (which depend on $P^{\text{Gibbs}}$, see Eq. 3), we set $\beta = 3$. For $a^{\text{EIG-O}}$ and $a^{\text{EIG-P}}$, we set $|\mathcal{E}^{\text{EIG}}| = 20N$ (see Appendix B.2 for details). For $a^{\text{JEIG}}$ we set $m = 5$, $n = 50$, $|\mathcal{D}_i| = \lceil |\mathcal{E}|/50 \rceil$ (i.e., 2% of all pairs) and each $\mathcal{D}_i$ is selected to contain pairs with large entropy (see Appendix B.3 for details). Finally, see Appendix C.7 for a detailed analysis and discussion of all hyperparameters.

**Performance evaluation.** In each iteration of the active CC procedure (Alg. 1), we calculate the Adjusted Rand Index (ARI) between the current clustering $c^i$ and the ground truth clustering $c^*$ (i.e., ground truth labels of dataset). Intuitively, ARI measures how similar the two clusterings are, where a value of 1 indicates they are identical. In Appendix C.4, we report the performance w.r.t. other evaluation metrics. Each active learning procedure is repeated 10 times with different random seeds, where the standard deviation is indicated by a shaded color or an error bar.

### 4.2 RESULTS

Figures 1-2 show the results for different datasets for oracle 1 and oracle 4, respectively. In Appendix C.4, we include the results for all oracles w.r.t. different performance metrics. We observe that all the information-theoretic acquisition functions introduced in this paper significantly outperform the baseline methods. In addition, the acquisition functions based on information gain ($a^{\text{EIG-O}}$, $a^{\text{EIG-P}}$ and $a^{\text{JEIG}}$) consistently outperform $a^{\text{Entropy}}$. This indicates the effectiveness of augmenting the similarity matrix $S$ with pseudo-similarities predicted by the current model $Q$ as $S_{uv} \sim P(\text{E}_{uv} \mid Q)$, before quantifying the model uncertainty. The acquisition functions based on information gain perform rather similarly. This is due to the fact that all of them are based on closely connected quantities (as described in Section 3.3). However, $a^{\text{JEIG}}$ is consistently among the best performing acquisition functions, while also being more computationally efficient compared to $a^{\text{EIG-O}}$ and $a^{\text{EIG-P}}$ (see Appendix C.6 for an investigation of the runtimes of all methods). Because of this, we exclude $a^{\text{EIG-O}}$ and $a^{\text{EIG-P}}$ in some cases due to their computational inefficiency. Both *maxmin* and *maxexp* perform significantly worse. This is likely because they spend too many queries with the goal of

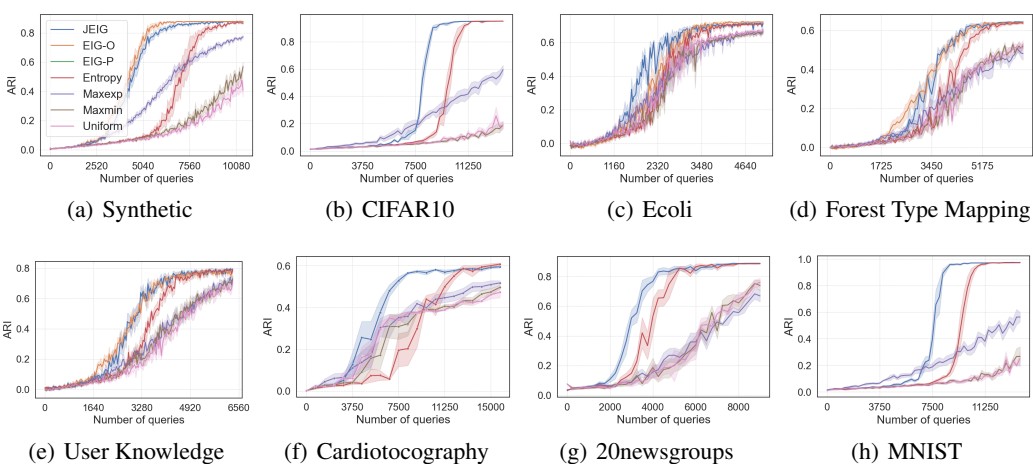

(a) Synthetic     (b) CIFAR10     (c) Ecoli     (d) Forest Type Mapping

(e) User Knowledge     (f) Cardiotocography     (g) 20newsgroups     (h) MNIST

Figure 2: Results for all datasets for oracle 4.

resolving the inconsistency of $S$ (see Appendix C.1 for details). However, the CC algorithm used (described in Appendix D) is robust to inconsistency in $S$. Finally, in Appendix C.5 we report our experiments on a small synthetic dataset with $N = 70$ objects using a small batch size $B = 5$. The purpose of this experiment is to further illustrate the benefit of the acquisition functions based on information gain, when differences due to batch diversity are eliminated.

### 4.3 SENSITIVITY ANALYSIS

We here investigate the sensitivity of acquisition functions when varying the noise level and batch size. All the results in this section are performed on the synthetic dataset using oracle 1. Figures 3(a)-3(b) show the results when varying the noise level $\gamma$ and batch size $B$, respectively. The $y$-axis corresponds to the area under the curve (AUC) of the active learning plot w.r.t. the respective performance metric (i.e., ARI) where higher is better. We see that our acquisition functions are very robust to noise. In addition, the benefit of our proposed acquisition functions increases with larger noise levels. This is consistent with previous work on active learning,

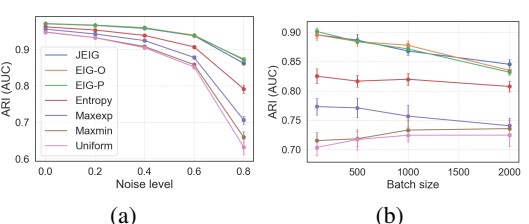

(a)         (b)

Figure 3: Results on the synthetic dataset when varying the noise level $\gamma$ and the batch size $B$.

where the benefit of many acquisition functions over uniform selection increases as the complexity of the problem increases. Expectedly, the performance decreases slightly as the batch size increases. However, the performance of our acquisition functions remains good even with large batch sizes.

## 5 CONCLUSION

In this paper, we proposed four effective information-theoretic acquisition functions to be used for active CC: $a^{\text{Entropy}}$, $a^{\text{EIG-O}}$, $a^{\text{EIG-P}}$ and $a^{\text{JEIG}}$. All of our methods significantly outperform the baseline methods by utilizing *model uncertainty*. We investigated the effectiveness of these methods via extensive experimental studies. The acquisition functions based on information gain ($a^{\text{EIG-O}}$, $a^{\text{EIG-P}}$ and $a^{\text{JEIG}}$) were consistently the best performing, where $a^{\text{JEIG}}$ has the benefit of being more computationally efficient.

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

## A  PROOFS

**Proposition 2.1.** *Eq. 1 can be simplified to $R^{CC}(\boldsymbol{c} \mid \boldsymbol{S}) = -\sum_{\substack{(u,v) \in \mathcal{E} \\ c_u = c_v}} S_{uv} + constant$, where the constant is independent of different clustering solutions (Chehreghani, 2013).*

*Proof.* As described in (Chehreghani, 2013; 2023), we can write the cost function in Eq. 1 as

$$
\begin{aligned}
R^{\mathrm{CC}}(\boldsymbol{c} \mid \boldsymbol{S}) &= \sum_{(u,v) \in \mathcal{E}} V(u, v \mid \boldsymbol{S}, \boldsymbol{c}) \\
&= \sum_{\substack{(u,v) \in \mathcal{E} \\ c_u = c_v}} \frac{1}{2}(|S_{uv}| - S_{uv}) + \sum_{\substack{(u,v) \in \mathcal{E} \\ c_u \neq c_v}} \frac{1}{2}(|S_{uv}| + S_{uv}) \\
&= \frac{1}{2} \sum_{(u,v) \in \mathcal{E}} |S_{uv}| - \frac{1}{2} \sum_{\substack{(u,v) \in \mathcal{E} \\ c_u = c_v}} S_{uv} + \frac{1}{2} \sum_{(u,v) \in \mathcal{E}} S_{uv} - \frac{1}{2} \sum_{\substack{(u,v) \in \mathcal{E} \\ c_u = c_v}} S_{uv} \\
&= \underbrace{\frac{1}{2} \sum_{(u,v) \in \mathcal{E}} (|S_{uv}| + S_{uv})}_{\text{constant}} - \sum_{\substack{(u,v) \in \mathcal{E} \\ c_u = c_v}} S_{uv}.
\end{aligned}
\tag{18}
$$

The first term in Eq. 18 is *constant* w.r.t. the choice of a particular clustering $\boldsymbol{c}$.

$\square$

**Theorem 3.1.** *Let $\ell : \mathbb{N} \to \mathcal{V}$ denote an object visitation schedule, which satisfies $\lim_{T \to \infty} |\{t \leq T : \ell(t) = u\}| = \infty, \forall u \in \mathcal{V}$. For arbitrary initial conditions, the asynchronous update rules defined by*

$$
Q_{uk}^{(t+1)} = \exp(-\beta M_{uk}^{(t)}) / \sum_{k' \in \mathbb{K}} \exp(-\beta M_{uk'}^{(t)}),
\tag{5}
$$

$$
M_{uk}^{(t+1)} = -\sum_{\substack{v \in \mathcal{V} \\ v \neq u}} S_{uv} Q_{vk}^{(t+1)},
\tag{6}
$$

*where $u = \ell(t)$, converge to a local minimum of Eq. 4.*

*Proof.* Given our cost function $R^{\mathrm{MC}}$ (Eq. 2), the *generalized free energy* is defined as (Hofmann et al., 1998)

$$
\begin{aligned}
\mathcal{F}_\beta(P) &\triangleq \mathbb{E}_{P(\mathbf{y})}[R^{\mathrm{MC}}(\mathbf{y})] - \frac{1}{\beta} H(P) \\
&= \sum_{\boldsymbol{c} \in \mathcal{C}} P(\boldsymbol{c}) R^{\mathrm{MC}}(\boldsymbol{c}) + \frac{1}{\beta} \sum_{\boldsymbol{c} \in \mathcal{C}} P(\boldsymbol{c}) \log P(\boldsymbol{c}),
\end{aligned}
\tag{19}
$$

for some $P \in \mathcal{P}$ where $\mathcal{P}$ is the set of distributions with sample space $\mathcal{C}$. The Gibbs distribution $P^{\mathrm{Gibbs}}$ minimizes the generalized free energy (Hofmann et al., 1998) and is called the *free energy*. It can be written as

$$
\mathcal{F}_\beta(P^{\mathrm{Gibbs}}) = -\frac{1}{\beta} \log \mathcal{Z},
\tag{20}
$$

where $\mathcal{Z} \triangleq \sum_{\boldsymbol{c}' \in \mathcal{C}} \exp(-\beta R^{\mathrm{MC}}(\boldsymbol{c}'))$ is the normalizing constant of the Gibbs distribution in Eq. 3. Given this, we can now simplify the KL-divergence.

$$
\begin{aligned}
D_{\mathrm{KL}}(Q \| P^{\mathrm{Gibbs}}) &= \sum_{c \in \mathcal{C}} Q(c) \log \frac{Q(c)}{P^{\mathrm{Gibbs}}(c)} \\
&= \sum_{c \in \mathcal{C}} Q(c) \log \frac{Q(c)}{\exp\left(-\beta \left(R^{\mathrm{MC}}(c) - \mathcal{F}_\beta(P^{\mathrm{Gibbs}})\right)\right)} \\
&= \sum_{c \in \mathcal{C}} Q(c) \left[\log Q(c) + \beta \left(R^{\mathrm{MC}}(c) - \mathcal{F}_\beta(P^{\mathrm{Gibbs}})\right)\right] \\
&= \sum_{u \in \mathcal{V}} \sum_{k \in \mathbb{K}} Q_{uk} \log Q_{uk} + \beta \mathbb{E}_{Q(\mathbf{y})}[R^{\mathrm{MC}}(\mathbf{y})] - \beta \mathcal{F}_\beta(P^{\mathrm{Gibbs}}) \\
&= \beta \mathbb{E}_{Q(\mathbf{y})}[R^{\mathrm{MC}}(\mathbf{y})] - \sum_{u \in \mathcal{V}} H(\mathbf{y}_u) - \beta \mathcal{F}_\beta(P^{\mathrm{Gibbs}}) \\
&= \beta \mathcal{F}_\beta(Q) - \beta \mathcal{F}_\beta(P^{\mathrm{Gibbs}}) \\
&\geq 0,
\end{aligned}
\tag{21}
$$

where $H(\mathbf{y}_u) \triangleq -\sum_{k \in \mathbb{K}} Q_{uk} \log Q_{uk}$ is the entropy of $\mathbf{y}_u$. The last inequality is a property of the KL-divergence. From this, we have the bound

$$
\mathcal{F}_\beta(P^{\mathrm{Gibbs}}) \leq \mathcal{F}_\beta(Q),
\tag{22}
$$

and minimizing the KL-divergence corresponds to minimizing the generalized free energy $\mathcal{F}_\beta$ w.r.t. factorial distributions $Q \in \mathcal{Q}$, which is consistent with the maximum entropy principle. From this, minimizing the KL-divergence corresponds to the following optimization problem.

$$
\begin{aligned}
Q^* &= \operatorname*{arg\,min}_{Q \in \mathcal{Q}} \mathcal{F}_\beta(Q) \\
\text{s.t.} \quad & \sum_{k \in \mathbb{K}} Q_{uk} = 1 \quad \forall u \in \mathcal{V}.
\end{aligned}
\tag{23}
$$

Then, by applying a Lagrangian relaxation to the constraint in Eq. 23 and setting the gradient of the objective w.r.t. $Q_{uk}$ to zero, we obtain

$$
\begin{aligned}
0 &= \frac{\partial}{\partial Q_{uk}} \left[\mathbb{E}_{Q(\mathbf{y})}[R^{\mathrm{MC}}(\mathbf{y})] - \frac{1}{\beta} \sum_{v \in \mathcal{V}} H(\mathbf{y}_v) + \sum_{w \in \mathcal{V}} \mu_w \left(\sum_{k \in \mathbb{K}} Q_{wk} - 1\right)\right] \\
&= \frac{\partial}{\partial Q_{uk}} \left[\sum_{c \in \mathcal{C}} \prod_{v \in \mathcal{V}} Q_{vc_v} R^{\mathrm{MC}}(c) - \frac{1}{\beta} \sum_{v \in \mathcal{V}} H(\mathbf{y}_v) + \sum_{w \in \mathcal{V}} \mu_w \left(\sum_{k \in \mathbb{K}} Q_{wk} - 1\right)\right] \\
&= \sum_{c \in \mathcal{C}} \prod_{\substack{v \in \mathcal{V} \\ v \neq u}} Q_{vc_v} \mathbf{1}_{\{c_u=k\}} R^{\mathrm{MC}}(c) + \frac{1}{\beta} \left(\log Q_{uk} + 1\right) + \mu_u \\
&= \mathbb{E}_{Q(\mathbf{y}|y_u=k)}[R^{\mathrm{MC}}(\mathbf{y})] + \frac{1}{\beta} \left(\log Q_{uk} + 1\right) + \mu_u,
\end{aligned}
\tag{24}
$$

where $\mu_u$'s are the Lagrange multipliers and we define $M_{uk} \triangleq \mathbb{E}_{Q(\mathbf{y}|y_u=k)}[R^{\mathrm{MC}}(\mathbf{y})]$ as the mean-fields, which correspond to the expected cost subject to the constraint that object $u$ is assigned to cluster $k$. We can simplify

$$
\begin{aligned}
M_{uk} &= \mathbb{E}_{Q(\mathbf{y}|y_u=k)}[R^{\mathrm{MC}}(\mathbf{y})] \\[2mm]
&= \mathbb{E}_{Q(\mathbf{y}|y_u=k)}\left[-\sum_{(v,w)\in\mathcal{E}} S_{vw}\right] \\[2mm]
&= \mathbb{E}_{Q(\mathbf{y}|y_u=k)}\left[-\sum_{l\in\mathbb{K}}\sum_{(v,w)\in\mathcal{E}} \mathbf{1}_{\{y_v=l\}}\mathbf{1}_{\{y_w=l\}}S_{vw}\right] \\[2mm]
&= -\sum_{l\in\mathbb{K}}\sum_{(v,w)\in\mathcal{E}} \mathbb{E}_{Q(\mathbf{y}|y_u=k)}[\mathbf{1}_{\{y_v=l\}}\mathbf{1}_{\{y_w=l\}}]S_{vw} \\[2mm]
&= -\sum_{l\in\mathbb{K}}\sum_{(v,w)\in\mathcal{E}} S_{vw}Q_{vl}Q_{wl} \\[2mm]
&= -\sum_{l\in\mathbb{K}}\sum_{\substack{v\in\mathcal{V}\\v\neq u}} S_{uv}Q_{ul}Q_{vl} - \sum_{l\in\mathbb{K}}\sum_{\substack{(v,w)\in\mathcal{E}\\v\neq u\\w\neq u}} S_{vw}Q_{vl}Q_{wl} \\[2mm]
&= -\sum_{\substack{v\in\mathcal{V}\\v\neq u}} S_{uv}Q_{vk} - \underbrace{\sum_{l\in\mathbb{K}}\sum_{\substack{(v,w)\in\mathcal{E}\\v\neq u\\w\neq u}} S_{vw}Q_{vl}Q_{wl}}_{\text{constant}}
\end{aligned}
\tag{25}
$$

where the last equality uses that $Q_{ul}=1$ if $l=k$ and 0 otherwise, according to $Q(\mathbf{c}\mid c_u=k)$. The second term of the last expression is a constant w.r.t. $Q_{uk}$ and is thus irrelevant for optimization (since it does not depend on $u$).

With the definition of $M_{uk}$, we can rewrite Eq. 24 as

$$
0 = M_{uk} + \frac{1}{\beta}\left(\log Q_{uk} + 1\right) + \mu_u.
\tag{26}
$$

Then, we have

$$
\begin{aligned}
\log Q_{uk} &= -\beta M_{uk} - \beta\mu_u \\
\Rightarrow Q_{uk} &= \exp\left(-\beta M_{uk}\right)\exp\left(-\beta\mu_u\right).
\end{aligned}
\tag{27}
$$

On the other hand, we have: $\sum_{k'} Q_{uk'} = 1$. Therefore,

$$
\begin{aligned}
\sum_{k'}\log Q_{uk'} &= \sum_{k'}\exp\left(-\beta M_{uk'}\right)\exp\left(-\beta\mu_u\right) = 1 \\
\Rightarrow \exp\left(-\beta\mu_u\right) &= \frac{1}{\sum_{k'}\exp\left(-\beta M_{uk'}\right)}.
\end{aligned}
\tag{28}
$$

Then, inserting Eq. 28 into Eq. 27 yields

$$
Q_{uk} = \frac{\exp\left(-\beta M_{uk}\right)}{\sum_{k'}\exp\left(-\beta M_{uk'}\right)}.
\tag{29}
$$

This derivation suggest an EM-type procedure for minimizing the KL-divergence $D_{\mathrm{KL}}(Q\|P^{\mathrm{Gibbs}})$, which consists of alternating between estimating $Q_{uk}$'s given $M_{uk}$'s and then updating $M_{uk}$'s given the new values of $Q_{uk}$'s (as described in Alg. 2).

Finally, we can compute the Hessian of the objective as

$$\begin{aligned}
\frac{\partial^2}{\partial Q_{uk}^2} \mathcal{F}_\beta(Q) &= \frac{\partial}{\partial Q_{uk}} M_{uk} + \frac{1}{\beta}(\log Q_{uk} + 1) + \mu_u \\
&= \frac{1}{\beta Q_{uk}} \\
&> 0.
\end{aligned} \tag{30}$$

The positivity of the Hessian in Eq. 30 ensures that the generalized free energy $\mathcal{F}_\beta(Q)$ is convex with respect to $Q_{uk}$ for each object $u$, guaranteeing that the update for $Q_{uk}$ strictly decreases $\mathcal{F}_\beta(Q)$ unless it is already at a local minimum. Since $\mathcal{F}_\beta(Q)$ is bounded from below by $\mathcal{F}_\beta(P^{\text{Gibbs}})$ and each object $u$ is updated infinitely often according to the object visitation schedule, the algorithm converges to a local minimum of the generalized free energy $\mathcal{F}_\beta$ within the space of factorial distributions $\mathcal{Q}$.

$\square$

# B ADDITIONAL DETAILS ABOUT INFORMATION-THEORETIC ACQUISITION FUNCTIONS

## B.1 DETAILED DERIVATION OF ENTROPY

Here we show a detailed derivation of the probability $P(E_{uv})$, which is used for the acquisition function based on entropy in Eq. 8. We have

$$\begin{aligned}
P(E_{uv} = 1) &= \mathbb{E}_{P^{\text{Gibbs}}(\mathbf{y})}[\mathbf{1}_{\{y_u = y_v\}}] \\
&\approx \mathbb{E}_{Q(\mathbf{y})}[\mathbf{1}_{\{y_u = y_v\}}] \\
&= \sum_{k' \in \mathbb{K}} Q_{uk'} \sum_{k'' \in \mathbb{K}} Q_{uk''} \mathbf{1}_{\{c_u = c_v\}} \\
&= \sum_{k \in \mathbb{K}} Q_{uk} Q_{vk} + \underbrace{\sum_{k' \in \mathbb{K}} \sum_{\substack{k'' \in \mathbb{K} \\ k'' \neq k'}} Q_{uk'} Q_{vk''} \mathbf{1}_{\{c_u = c_v\}}}_{=0} \\
&= \sum_{k \in \mathbb{K}} Q_{uk} Q_{vk}.
\end{aligned} \tag{31}$$

One can also show that $P(E_{uv} = -1) \approx \mathbb{E}_{Q(\mathbf{c})}[\mathbf{1}_{\{c_u \neq c_v\}}(\mathbf{c})]$ which can be simplified to $P(E_{uv} = -1) = \sum_{k,k' \in \mathbb{K}} Q_{uk} Q_{vk'} \mathbf{1}_{\{k \neq k'\}} = 1 - P(E_{uv} = 1)$.

## B.2 DERIVATIONS OF EIG

In this section, we include a detailed derivation of the acquisition function defined in Eq. 13. In addition, we derive the acquisition function $a^{\text{EIG-P}}$.

$$H(\mathbf{y}) = -\sum_{\mathbf{c} \in \mathcal{C}} P^{\text{Gibbs}}(\mathbf{y} = \mathbf{c}) \log P^{\text{Gibbs}}(\mathbf{y} = \mathbf{c}). \tag{32}$$

A mean-field approximation $Q(\mathbf{y} = \mathbf{c}) = \prod_{u=1}^{N} Q(y_u = c_u)$ of $P^{\text{Gibbs}}$ assumes independence between each cluster label $y_u$. This means we have

$$H(\mathbf{y}) = -\sum_{\boldsymbol{c}\in\mathcal{C}} Q(\mathbf{y}=\boldsymbol{c})\log Q(\mathbf{y}=\boldsymbol{c})$$

$$= -\sum_{\boldsymbol{c}\in\mathcal{C}} \prod_{u=1}^{N} Q(\mathbf{y}_u=c_u)\log\prod_{v=1}^{N} Q(\mathbf{y}_v=c_v)$$

$$= -\sum_{\boldsymbol{c}\in\mathcal{C}} \prod_{u=1}^{N} Q(\mathbf{y}_u=c_u)(\sum_{v=1}^{N}\log Q(\mathbf{y}_v=c_v))$$

$$= -\sum_{\boldsymbol{c}\in\mathcal{C}} \sum_{v=1}^{N}\prod_{u=1}^{N} Q(\mathbf{y}_u=c_u)\log Q(\mathbf{y}_v=c_v)$$

$$= -\sum_{v=1}^{N}\sum_{k\in\mathbb{K}} \sum_{\substack{\boldsymbol{c}\in\mathcal{C}\\c_v=k}}\prod_{u=1}^{N} Q(\mathbf{y}_u=c_u)\log Q(\mathbf{y}_v=c_v) \tag{33}$$

$$= -\sum_{v=1}^{N}\sum_{k\in\mathbb{K}} \log Q(\mathbf{y}_v=c_v)\sum_{\substack{\boldsymbol{c}\in\mathcal{C}\\c_v=k}}\prod_{u=1}^{N} Q(\mathbf{y}_u=c_u)$$

$$= -\sum_{v=1}^{N}\sum_{k\in\mathbb{K}} \log Q(\mathbf{y}_v=c_v)Q(\mathbf{y}_v=c_v)\underbrace{\left(\sum_{\substack{\boldsymbol{c}\in\mathcal{C}\\c_v=k}}\prod_{\substack{u=1\\u\neq v}}^{N} Q(\mathbf{y}_u=c_u)\right)}_{=1}$$

$$= -\sum_{v=1}^{N}\sum_{k\in\mathbb{K}} Q(\mathbf{y}_v=c_v)\log Q(\mathbf{y}_v=c_v)$$

$$= \sum_{v=1}^{N} H(\mathbf{y}_v\mid\boldsymbol{Q}).$$

Furthermore, we assume independence between pairs $\mathbf{E}$ given a mean-field approximation $\boldsymbol{Q}$. Consequently, we have $P(\mathbf{E}) = \prod_{(w,l)\in\mathcal{E}} P(\mathrm{E}_{wl}\mid\boldsymbol{Q})$. Then, one can derive $H(\mathbf{E}) = \sum_{(w,l)\in\mathcal{E}} H(\mathrm{E}_{wl}\mid\boldsymbol{Q})$ following the same derivation as shown in Eq. 33. In addition, we choose to approximate $H(\mathbf{y}\mid\mathrm{E}_{uv}=e)$ and $H(\mathbf{E}\mid\mathrm{E}_{uv}=e)$ using conditional mean-field approximation $\boldsymbol{Q}^{(S_{uv}=e)}$. As a consequence, the joint conditional entropies reduces to the sum of entropies over individual variables. This is shown following the same derivation shown in Eq. 33.

From this, we obtain the acquisition function $a^{\mathrm{EIG\text{-}O}}$ defined in Eq. 13 of the main paper. In addition, we define an acquisition function which computes the *expected information gain over the clustering relation of pairs* (EIG-P).

$$a^{\mathrm{EIG\text{-}P}}(u,v) \triangleq \sum_{(w,l)\in\mathcal{E}} H(\mathrm{E}_{wl}\mid\boldsymbol{Q}) - \sum_{e\in\{-1,+1\}} P(\mathrm{E}_{uv}=e\mid\boldsymbol{Q})H(\mathrm{E}_{wl}\mid\boldsymbol{Q}^{(S_{uv}=e)}). \tag{34}$$

$a^{\mathrm{EIG\text{-}P}}$ is also computed using Alg. 3.

### B.3 JEIG ALGORITHM

Alg. 4 outlines how the acquisition function $a^{\mathrm{JEIG}}$ (Eq. 17) is calculated. The algorithm begins by initializing $\boldsymbol{Q}$ and $\boldsymbol{M}$ by running Algorithm 2. Then, the algorithm loops $m$ times. In each of the $m$ iterations, a subset $\mathcal{D}_i\subseteq\mathcal{E}$ is selected. Then, lines 7-10 computes a Monte-Carlo estimation of the expectation $\mathbb{E}_{\boldsymbol{e}\sim P(\mathbf{E}_{\mathcal{D}_i})}[H(\mathrm{E}_{uv}\mid\mathbf{E}_{\mathcal{D}_i}=\boldsymbol{e})]$. We found that the selection of $\mathcal{D}_i$ (on line 6) can be done in a number of ways, with good performance. In the experiments of this paper, we select the top-$|\mathcal{D}_i|$ pairs according to $\log(a^{\mathrm{Entropy}}(u,v)) + \epsilon_{uv}$ where $\epsilon_{uv}\sim\mathrm{Gumbel}(0;1)$. In other words, the top-$|\mathcal{D}_i|$ pairs according to $a^{\mathrm{Entropy}}$ with some added acquisition noise (as explained in Section 4.1).

This leads to diversity among the selected $\mathcal{D}_i$, while containing pairs with large entropy. Pairs with large entropy are likely to have large impact on each $\mathrm{E}_{uv}$, and are therefore important to include in $\mathcal{D}_i$. We set $|\mathcal{D}_i| = 0.02|\mathcal{E}|$ (i.e., 2% of all pairs), $m = 5$ and $n = 50$ for all datasets. See Appendix C.7 for more details about this.

---

**Algorithm 4** JEIG

---

1: **Input:** Similarity matrix $\boldsymbol{S}$, current clustering $\boldsymbol{c}^i$, concentration parameter $\beta$.
2: $M_{uk} \leftarrow -\sum_{v:c_v^i=k} S_{uv}, \forall u \in \mathcal{V}, \forall k \in \mathbb{K}$
3: $\boldsymbol{Q}, \boldsymbol{M} \leftarrow \text{MeanField}(\boldsymbol{S}, \boldsymbol{M}, \beta)$
4: $a^{\text{JEIG}}(u, v) \leftarrow 0 \quad \forall (u, v) \in \mathcal{E}$
5: **for** $i \leftarrow 1$ to $m$ **do**
6: $\quad \mathcal{D}_i \leftarrow \text{SelectPairs}(\mathcal{E})$ $\qquad\qquad\qquad\qquad\qquad\qquad \triangleright \mathcal{D}_i \subseteq \mathcal{E}$
7: $\quad$ **for** $j \leftarrow 1$ to $n$ **do**
8: $\qquad \boldsymbol{e} \sim P(\mathbf{E}_{\mathcal{D}_i})$
9: $\qquad \boldsymbol{Q}^{(\boldsymbol{S}_{\mathcal{D}_i}=\boldsymbol{e})} \leftarrow \text{MeanField}(\boldsymbol{S}, \boldsymbol{M}, \beta \mid \boldsymbol{S}_{\mathcal{D}_i} = \boldsymbol{e})$
10: $\qquad a^{\text{JEIG}}(u, v) \leftarrow a^{\text{JEIG}}(u, v) + H(\mathrm{E}_{uv} \mid \boldsymbol{Q}^{(\boldsymbol{S}_{\mathcal{D}_i}=\boldsymbol{e})})/n \quad \forall (u, v) \in \mathcal{E}$
11: $\quad$ **end for**
12: **end for**
13: $a^{\text{JEIG}}(u, v) \leftarrow H(\mathrm{E}_{uv} \mid \boldsymbol{Q}) - a^{\text{JEIG}}(u, v)/m \quad \forall (u, v) \in \mathcal{E}$
14: **return** $a^{\text{JEIG}}$

---

## C EXPERIMENTS: MORE DETAILS AND FURTHER RESULTS

In this section, we describe the datasets in more detail and provide further experimental results. The experimental settings are identical to Section 4, unless otherwise specified.

### C.1 MAXMIN AND MAXEXP

In this section, we explain the acquisition functions *maxmin* and *maxexp* introduced in (Aronsson & Chehreghani, 2024). First, the transitive property implies if $S_{uv} \geq 0$ and $S_{uw} \geq 0$ then $S_{vw} \geq 0$ or if $S_{uv} \geq 0$ and $S_{uw} < 0$ then $S_{vw} < 0$. Then, assuming the ground-truth similarity matrix $\boldsymbol{S}^*$ is consistent (i.e., it does not violate transitive property) would imply that explicitly resolving (or preventing) the inconsistency in $\boldsymbol{S}$ may be informative. Both *maxmin* and *maxexp* are based on this idea.

Let $\mathcal{T}$ be the set of triples $(u, v, w)$ of distinct objects in $\mathcal{V}$, i.e., $|\mathcal{T}| = \binom{N}{3}$. Let $\mathcal{T}_{uv} = \{t \in \mathcal{T} \mid u, v \in t\}$ be the set of triples that include the pair $(u, v)$. Let $\mathcal{C}_t$ be the set of clustering solutions for the objects in the triple $t$. Finally, let $\mathcal{E}_t = \{(u, v) \in \mathcal{E} \mid u, v \in t\}$ be the set of pairs in the triple $t$, and $e_t = \arg\min_{(u,v) \in \mathcal{E}_t} |S_{uv}|$ is the pair in $\mathcal{E}_t$ with the smallest absolute similarity. Given this, *maxmin* is defined as[3]

$$a^{\text{Maxmin}}(u, v) \triangleq \max_{t \in \mathcal{T}_{uv}} \min_{\boldsymbol{c} \in \mathcal{C}_t} R(\boldsymbol{c} \mid \mathcal{E}_t) \mathbf{1}_{\{e_t = (u,v)\}}, \tag{35}$$

where $R(\boldsymbol{c} \mid \mathcal{E}_t) \triangleq \sum_{(u,v) \in \mathcal{E}_t} V(u, v \mid \boldsymbol{c})$. Intuitively, *maxmin* begins by ranking each triple according to how much inconsistency they induce (i.e., violation of transitive property). Then, from each of the top-$B$ triples $t$, the pair in $\mathcal{E}_t$ with smallest absolute similarity is selected (i.e., most uncertain according to its similarity). The goal is thus to reduce inconsistency by resolving violations of the transitive property in triples. In our experiments, we observe that this can be ineffective, likely due to robustness to inconsistency in $\boldsymbol{S}$ by the CC algorithm used. See discussion in experiments for more details. From Eq. 35 we see that *maxmin* quantifies the inconsistency by the cost of the best clustering in $\mathcal{C}_t$ (in short, this cost is non-zero for triples that violate the transitive property, and zero otherwise). The maximization over $\mathcal{T}_{uv}$ ensures the most violating triple that includes the pair $(u, v)$ is considered. *maxexp* works analogously to *maxmin* except the term $\min_{\boldsymbol{c} \in \mathcal{C}_t} R(\boldsymbol{c} \mid \mathcal{E}_t)$ is replaced by an expectation of the cost of all clustering solutions in $\mathcal{C}_t$.

---

[3]This formulation of *maxmin* is equivalent to Algorithm 3 of (Aronsson & Chehreghani, 2024), except the algorithm overcomes the computational issues of iterating all $\binom{N}{3}$ triples.

## C.2   DETAILS ABOUT ORACLE 4

In this section, we describe the details of Oracle 4. Given a dataset $\mathbf{X}$ and ground-truth labels $c^*$, the ground-truth similarities are defined as $S^*_{uv} = +1$ if $c^*_u = c^*_v$, and $-1$ otherwise. The dataset $\mathbf{X}$ is then split into two disjoint parts: $\mathbf{X} = \mathbf{X}_{\text{train}} \cup \mathbf{X}_{\text{test}}$, with 30% of the data allocated to the training set and 70% to the test set. Given this, the sizes of $\mathbf{X}_{\text{train}}$ and $\mathbf{X}_{\text{test}}$ are restricted to a maximum of 5000 and 1000 samples, respectively.

Next, we define a pairwise prediction model $f_\theta : \mathbf{X} \times \mathbf{X} \to [-1, +1]$. In our experiments, $f_\theta$ is a fully connected neural network with 6 hidden layers of sizes $[1024, 2048, 512, 248, 64]$, using ReLU activations. The input to the network is the concatenation of two feature vectors, i.e., $\mathbf{x}_u \oplus \mathbf{x}_v$. We treat this as a binary classification problem, where the output of the neural network represents the probability that the similarity between $u$ and $v$ is $+1$. Denoting this probability as $p_{uv}$, we transform it to a similarity score in $[-1, +1]$ using the transformation $2 \cdot p_{uv} - 1$. The network is trained using the standard cross-entropy loss function over 30 epochs.

We then construct a training dataset where the inputs are $\{\mathbf{x}_u \oplus \mathbf{x}_v\}_{\mathbf{x}_u, \mathbf{x}_v \in \mathbf{X}_{\text{train}}}$, and the corresponding labels are $S^*_{uv}$. In practice, we limit the number of training pairs to a maximum of 30,000, as the total number of possible pairs would otherwise be prohibitively large, resulting in extremely slow training.

Finally, the active correlation clustering experiments are conducted on the data points in $\mathbf{X}_{\text{test}}$. It is important to note that the ground-truth similarities of pairs in $\mathbf{X}_{\text{test}}$ are not used during the training of $f_\theta$.

## C.3   DESCRIPTION OF DATASETS

A detailed description of all eight datasets used is provided below. Datasets 2-6 are taken from the UCI machine learning repository (Kelly et al., 2023) (all of which are released under the CC BY 4.0 license).

1. **CIFAR10** (Krizhevsky, 2009). This dataset consists of 60000 $32 \times 32$ color images in 10 different classes (with 6000 images per class). A random subset of $N = 1000$ images (with $|\mathcal{E}| = 499, 500$) is used.[4] Cluster sizes: [91, 96, 107, 89, 99, 113, 96, 93, 112, 104]. We use a ResNet18 model (He et al., 2015) trained on the full CIFAR10 dataset in order to embed the 1000 images into a 512-dimensional space. For oracle 4, $f_\theta$ is trained on data points embedded into the latent space. We set $|\mathcal{E}^0| = 2500$. The batch size is set to $B = 1250$.

2. **20newsgroups**. This dataset consists of 18846 newsgroups posts (in the form of text) on 20 topics (clusters). We consider a subset of 5 topics: "rec.sport.baseball", "soc.religion.christian", "rec.autos", "talk.politics.mideast", "misc.forsale". We use a random sample of $N = 1000$ posts (with $|\mathcal{E}| = 499, 500$). Cluster sizes: [201, 190, 201, 217, 191]. We use the `distilbert-base-uncased` transformer model loaded from the Flair Python library (Akbik et al., 2018) in order to embed each of the 1000 documents (data points) into a 768-dimensional latent space. For oracle 4, $f_\theta$ is trained on data points embedded into the latent space. We set $|\mathcal{E}^0| = 2500$. The batch size is set to $B = 250$.

3. **Cardiotocography**. This dataset includes 2126 fetal cardiotocograms consisting of 22 features and 10 classes. We use a sample of $N = 1000$ data points (with $|\mathcal{E}| = 499, 500$). Cluster sizes: [180, 275, 27, 35, 31, 148, 114, 62, 28, 100]. We set $|\mathcal{E}^0| = 2500$. The batch size is set to $B = 750$.

4. **Ecoli**. This is a biological dataset on the cellular localization sites of 8 types (clusters) of proteins which includes $N = 336$ samples (with $|\mathcal{E}| = 56, 280$). Cluster sizes: [137, 76, 1, 2, 37, 26, 5, 52]. We set $|\mathcal{E}^0| = 280$. The batch size is set to $B = 85$.

5. **Forest Type Mapping**. This is a remote sensing dataset of $N = 523$ samples collected from forests in Japan and grouped in 4 different forest types (clusters) (with $|\mathcal{E}| = 136, 503$). Cluster sizes: [168, 84, 86, 185]. We set $|\mathcal{E}^0| = 500$. The batch size is set to $B = 350$.

---

[4]For oracles 1-3 we simply select $N = 1000$ random data points. For oracle 4 we obtain the random sample based on the construction of $\mathbf{X}_{\text{test}}$, as explained in the previous section. The same applies to the other datasets.

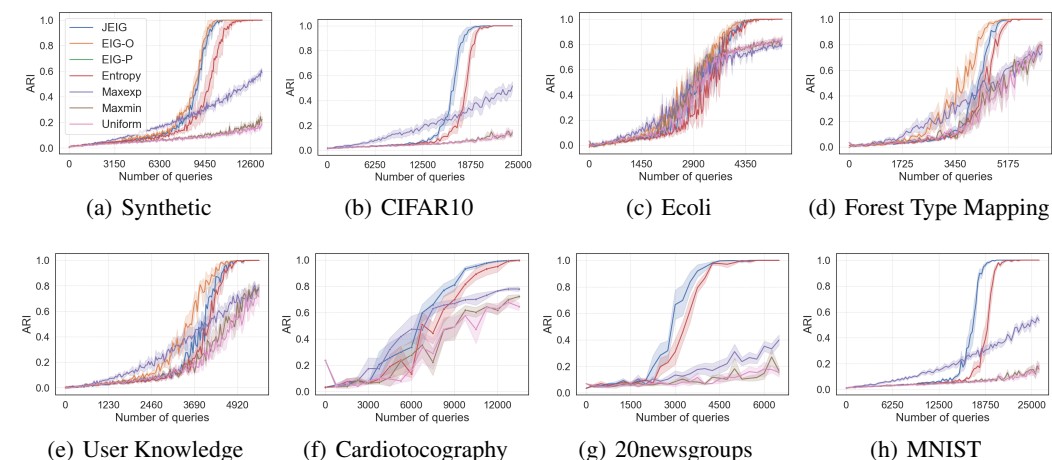

Figure 4: Results for oracle 2 with variance $\gamma = 1.3$. The evaluation metric is the adjusted rand index (ARI).

6. **User Knowledge Modelling**. This dataset contains 403 students' knowledge status on Electrical DC Machines grouped in 4 classes (with $|\mathcal{E}| = 81,003$). Cluster sizes: [111, 129, 116, 28, 19]. We set $|\mathcal{E}^0| = 400$. The batch size is set to $B = 200$.

7. **MNIST** (LeCun et al., 1998). This dataset consists of 60000 $28 \times 28$ grayscale images of handwritten digits. We use a sample of $N = 1000$ images (with $|\boldsymbol{E}| = 499500$). Cluster sizes: [105, 109, 111, 112, 104, 86, 99, 88, 88, 98]. We use a simple CNN model trained on the MNIST dataset in order to embed the 1000 images into a 128-dimensional space. For oracle 4, $f_\theta$ is trained on data points embedded into the latent space. We set $|\mathcal{E}^0| = 2500$. The batch size is set to $B = 1250$.

8. **Synthetic**. This is a synthetically generated dataset (normally distributed 10-dimensional data points) with $N = 500$ (and $|\mathcal{E}| = 124,750$) data points split evenly into 10 clusters. We set $|\mathcal{E}^0| = 500$. The batch size is set to $B = 300$.

## C.4 FURTHER RESULTS

Figures 4 and 5 show results for oracles 2 and 3, respectively, where the evaluation metric is the adjusted rand index (ARI). Figures 6-9 show results for all oracles, where the evaluation metric is the adjusted mutual information (AMI). All results are consistent with the insights from Figures 1-2 from the main paper, where all information-theoretic acquisition functions proposed in this paper outperform the baselines. In addition, we observe that the acquisition functions based on information gain consistently outperforms $a^{\text{Entropy}}$.

## C.5 SMALL BATCH SIZE

In Figure 10, we show results for oracle 1 for a synthetic dataset with $N = 70$ objects (and $|\mathcal{E}| = 2415$ pairs) using a batch size of $B = 5$. The noise level is $\gamma = 0.4$. In this experiment, *we do not use any acquisition noise* in order to improve batch diversity. The purpose of this experiment is to further illustrate the benefit of the acquisition functions based on information gain compared to entropy, when differences due to batch diversity are (mostly) removed. We observe that $a^{\text{EIG-O}}$, $a^{\text{EIG-P}}$ and $a^{\text{JEIG}}$ outperform $a^{\text{Entropy}}$.

## C.6 RUNTIME

Each active learning procedure was executed on 1 core of an Intel(R) Xeon(R) Gold 6338 CPU @ 2GHz (with 32 cores total). We have access to a compute cluster with many of these CPU's allowing us to execute many procedures in parallell. Each CPU has access to 128GB of RAM (shared among cores), but much less would suffice for our experiments.

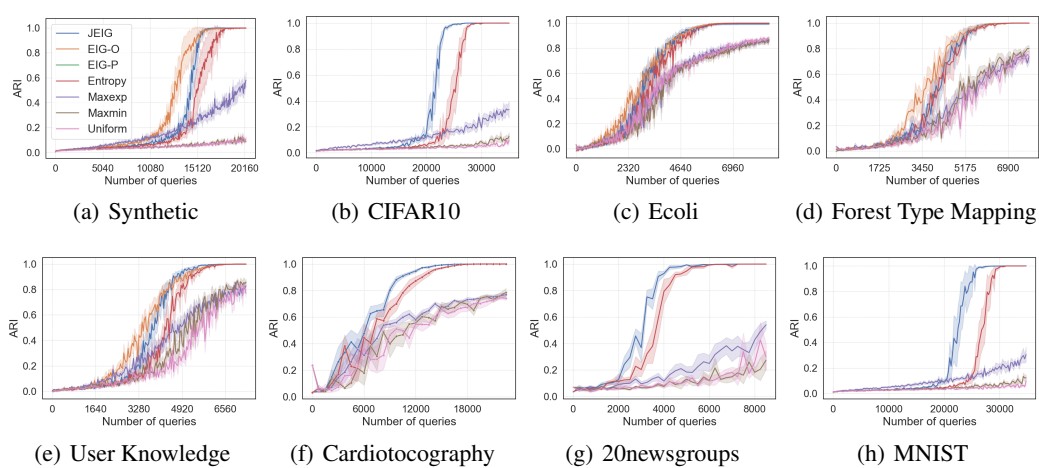

Figure 5: Results for oracle 3 with noise level $\gamma = 0.2$. The evaluation metric is the adjusted rand index (ARI).

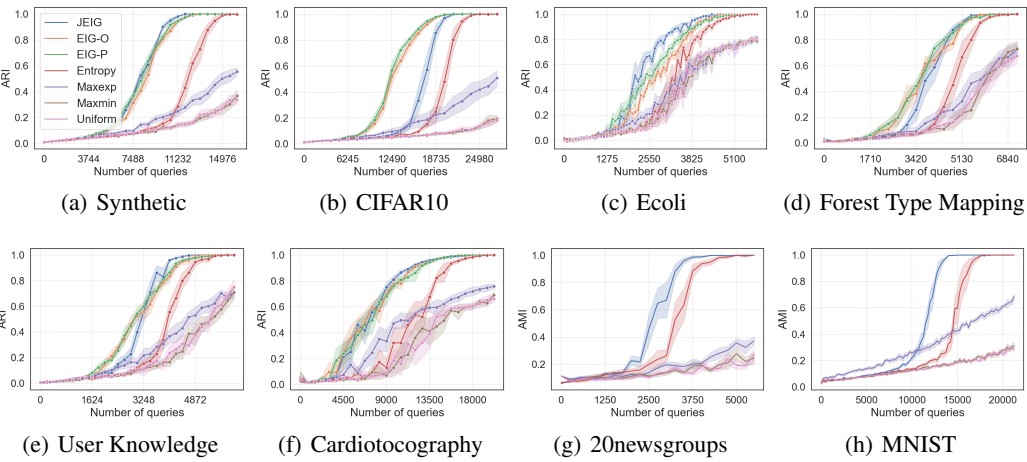

Figure 6: Results for oracle 1 with noise level $\gamma = 0.4$. The evaluation metric is the adjusted mutual information (AMI).

In Figure 11 we show the runtime of each iteration in seconds for all acquisition functions and datasets. We observe that $a^{\text{Entropy}}$ is very efficient (comparable to other baseline methods). In addition, we see that out of the acquisition functions based on information gain, $a^{\text{JEIG}}$ is the most efficient and is quite close to $a^{\text{Entropy}}$. Expectedly, $a^{\text{EIG-O}}$ and $a^{\text{EIG-P}}$ are the least efficient. This is because we run Alg. 2 numerous times (as discussed in Section B.2). Out of these two, $a^{\text{EIG-P}}$ is the most inefficient since it involves a sum over all $\binom{N}{2}$ pairs (see Eq. 34) in each iteration of Alg. 3.

## C.7 HYPERPARAMETERS

In this section, we present a detailed analysis of all hyperparameters. All experiments use oracle 1.

### C.7.1 EIG

In Figure 12, we show results for the acquisition functions $a^{\text{EIG-O}}$ (left) and $a^{\text{EIG-P}}$ (right) with different values of $|\mathcal{E}^{\text{EIG}}|$ (using oracle 1). See Alg. 3 for the usage of $\mathcal{E}^{\text{EIG}}$. We observe that the performance does not improve much beyond $|\mathcal{E}^{\text{EIG}}| = 10N$. This indicates both of these acquisition functions will perform well when evaluation Eq. 13 or Eq. 34 for $O(N)$ of the pairs (instead of all $O(N^2)$ pairs).

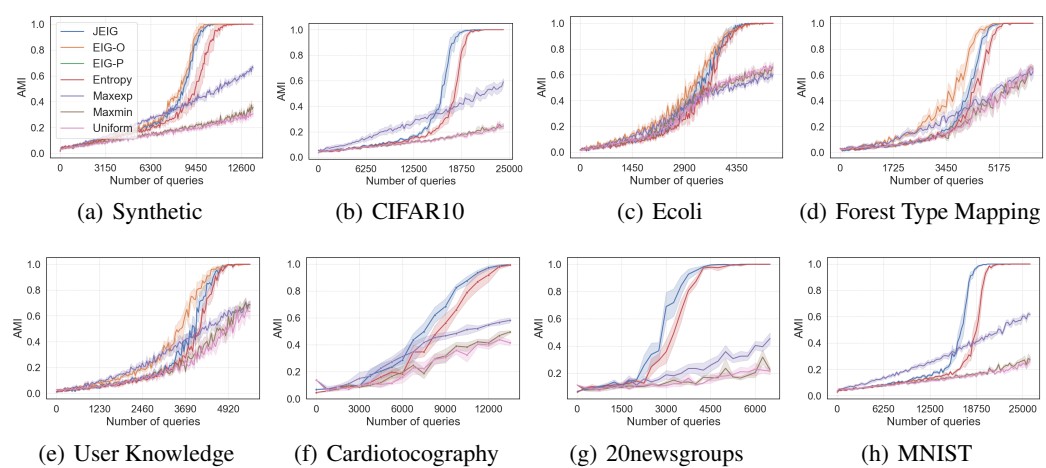

(a) Synthetic     (b) CIFAR10     (c) Ecoli     (d) Forest Type Mapping

(e) User Knowledge     (f) Cardiotocography     (g) 20newsgroups     (h) MNIST

Figure 7: Results for oracle 2 with variance $\gamma = 1.3$. The evaluation metric is the adjusted mutual information (AMI).

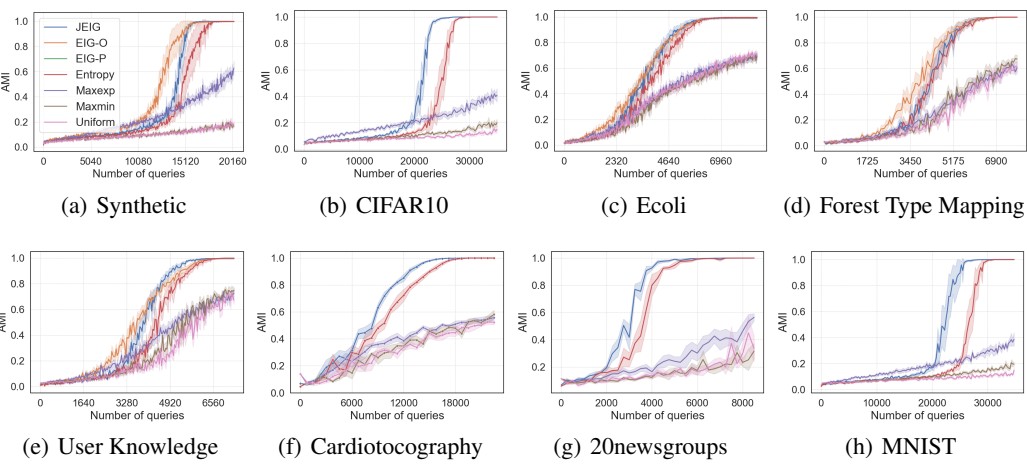

(a) Synthetic     (b) CIFAR10     (c) Ecoli     (d) Forest Type Mapping

(e) User Knowledge     (f) Cardiotocography     (g) 20newsgroups     (h) MNIST

Figure 8: Results for oracle 3 with noise level $\gamma = 0.2$. The evaluation metric is the adjusted mutual information (AMI).

### C.7.2 JEIG

In Figure 13, we show results for the acquisition function $a^{\text{JEIG}}$ when varying $m$, $n$ and $|\mathcal{D}_i|$. The left plot show results when varying $|\mathcal{D}_i|$ with $m$ and $n$ fixed to 5 and 50, respectively. As explained in Appendix B.3, each $\mathcal{D}_i$ is selected as the top-$|\mathcal{D}_i|$ pairs according to $\log(a^{\text{Entropy}}(u, v)) + \epsilon_{uv}$ where $\epsilon_{uv} \sim \text{Gumbel}(0; 1)$. The right plot show results when varying $m$ and $n$ with $|\mathcal{D}_i|$ fixed to $0.02|\mathcal{E}|$ (2% of all pairs). We observe that $|\mathcal{D}_i| = 0.02|\mathcal{E}|$ performs the best. A smaller value means we do not capture enough information about each $\mathbf{E}_{uv}$ and a too large value leads to exaggerated selection bias, as explained at the end of Section 3.3.3. We find $|\mathcal{D}_i| = 0.02|\mathcal{E}|$ to work well for all datasets considered in this paper. However, there may be other values that perform equally well (or better). Finally, we observe that larger values of $m$ and $n$ expectedly lead to better performance. A larger value of $m$ means we capture more information about each $\mathbf{E}_{uv}$. A larger value of $n$ means the Monte-Carlo estimation of the expectation $\mathbb{E}_{e \sim P(\mathbf{E}_{\mathcal{D}_i})}[H(\mathbf{E}_{uv} \mid \mathbf{E}_{\mathcal{D}_i} = e)]$ becomes more accurate.

### C.7.3 CONCENTRATION PARAMETER $\beta$

In Figure 14, we show results for the information-theoretic acquisition functions $a^{\text{Entropy}}$, $a^{\text{EIG-O}}$ and $a^{\text{JEIG}}$ when varying the hyperparameter $\beta$. The parameter $\beta$ is a concentration parameter used in

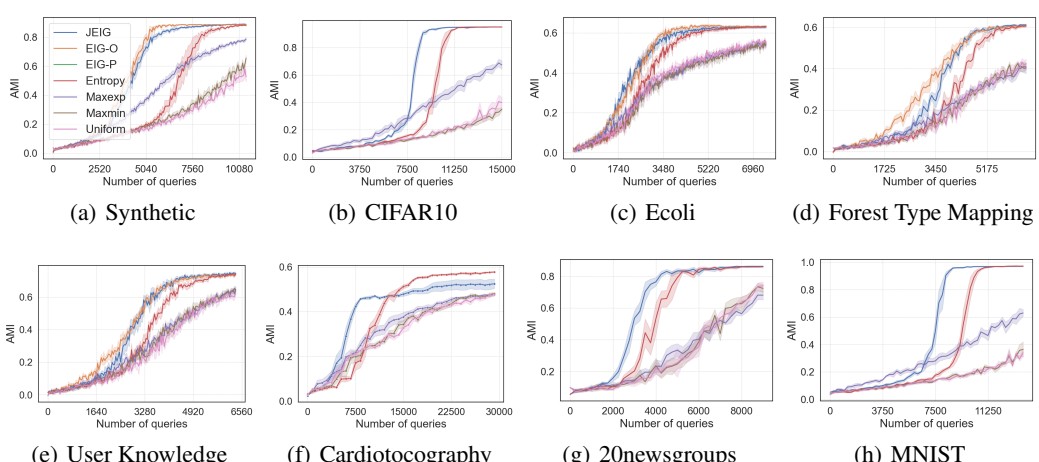

(a) Synthetic  (b) CIFAR10  (c) Ecoli  (d) Forest Type Mapping

(e) User Knowledge  (f) Cardiotocography  (g) 20newsgroups  (h) MNIST

Figure 9: Results for oracle 4. The evaluation metric is the adjusted mutual information (AMI).

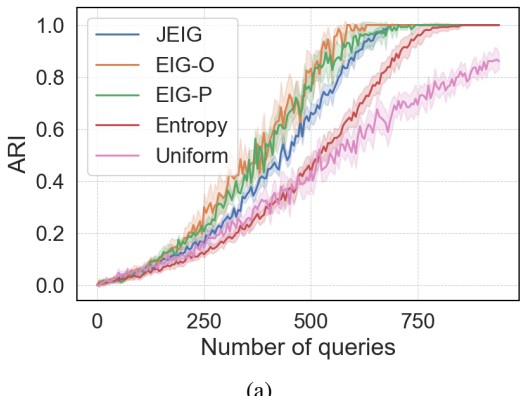

(a)

Figure 10: Results on synthetic dataset with $N = 70$ and $|\mathcal{E}| = 2415$ using a small batch size $B = 5$ without any acquisition noise. The noise level is $\gamma = 0.4$. This experiment used oracle 1.

the definition of the Gibbs distribution from Eq. 3. As a consequence, it is also used in Alg. 2 (mean-field), which is frequently used in this paper. In this setting, $\beta$ is the well-known inverse temperature of a Gibbs distribution (having this parameter with a Gibbs distribution is very common). A large $\beta$ will concentrate more probability mass on a cluster $k$ with larger cost $M_{uk}^t$. A smaller $\beta$ will make the probabilities $Q_{uk}^t$ more uniform across different clusters. $\beta$ may therefore have an impact on the resulting clustering (i.e., assignment probabilities $\boldsymbol{Q}$ which is used by all information-theoretic acquisition functions). See (Chehreghani et al., 2012) for more details about the impact of $\beta$. We observe that a value of $\beta = 3$ performs the best for all acquisition functions.

## C.8  UTILIZING FEATURES FOR ACTIVE CORRELATION CLUSTERING

The primary focus of this paper is on standard correlation clustering (Bansal et al., 2004; Bonchi et al., 2014) in the active learning setting, where no feature vectors are assumed to be available, i.e., similar to the recent works (Bressan et al., 2019; García-Soriano et al., 2020; Aronsson & Chehreghani, 2024; Kuroki et al., 2024). However, our framework is generic enough to also incorporate feature vectors when available for the objects. In this section, we propose an innovative but simple method. After line 6 of Alg. 1, we introduce a prediction component that predicts similarities based on the queries made so far. This component works as follows.

For all queried pairs $(u, v)$ (i.e., pairs where the oracle has provided the similarity $S_{uv}$), we concatenate their feature vectors $\mathbf{x}_u \oplus \mathbf{x}_v$ and add them to a dataset, using $S_{uv}$ as the corresponding label.

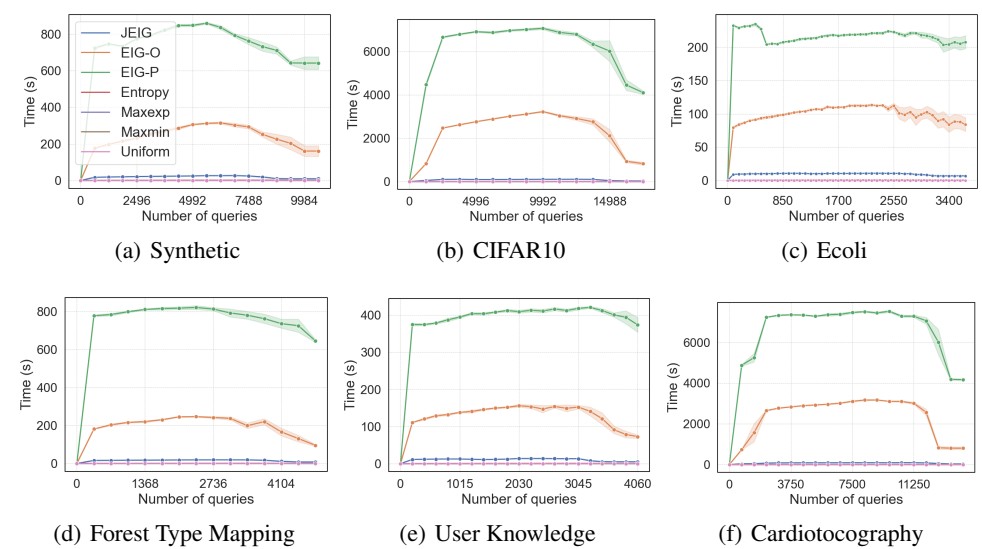

Figure 11: Runtime of all acquisition functions on all datasets with noise level $\gamma = 0.4$. The $y$-axis corresponds to the execution time in seconds of each iteration. This corresponds to the same experiments presented in Figure 1.

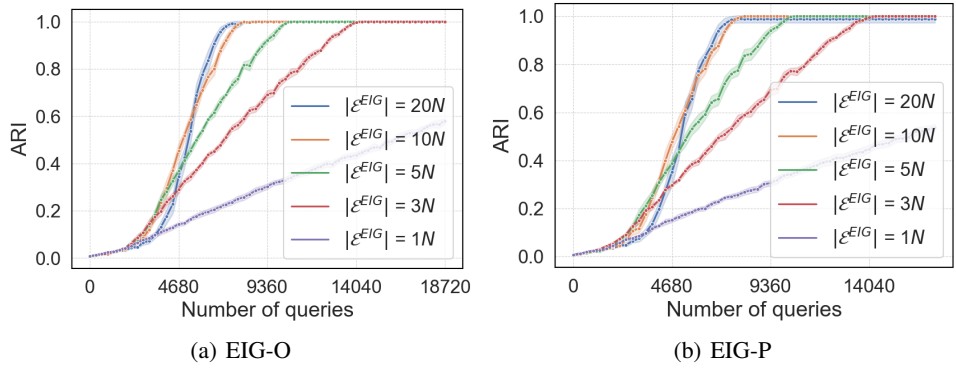

Figure 12: Results for acquisition functions $a^{\text{EIG-O}}$ (left) and $a^{\text{EIG-P}}$ (right) with different values of $|\mathcal{E}^{\text{EIG}}|$.

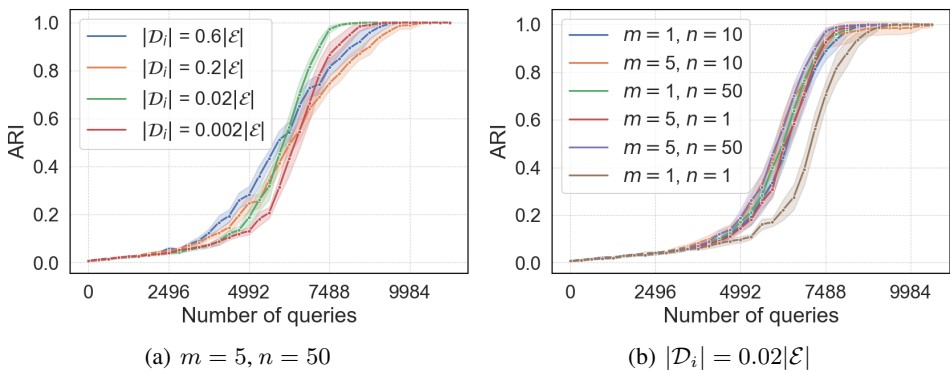

Figure 13: Results for acquisition function $a^{\text{JEIG}}$ when varying hyperparameters $m$, $n$ and $|\mathcal{D}_i|$.

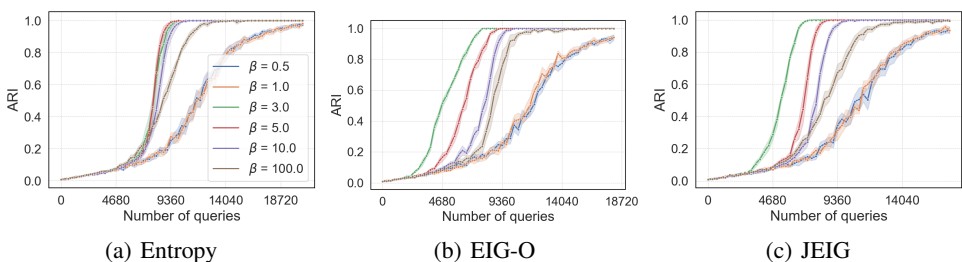

(a) Entropy             (b) EIG-O             (c) JEIG

Figure 14: Results of information-theoretic acquisition functions $a^{\text{Entropy}}$, $a^{\text{EIG-O}}$ and $a^{\text{JEIG}}$ when varying hyperparameter $\beta$ (used in Eq. 3)

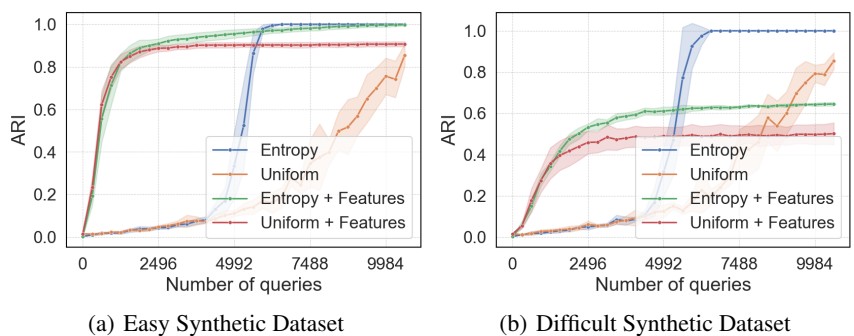

(a) Easy Synthetic Dataset          (b) Difficult Synthetic Dataset

Figure 15: Performance of $a^{\text{Entropy}}$ and $a^{\text{Uniform}}$ when leveraging feature vectors across two synthetic datasets, one with a simpler (easy) feature space and the other with a more complex (hard) feature space.

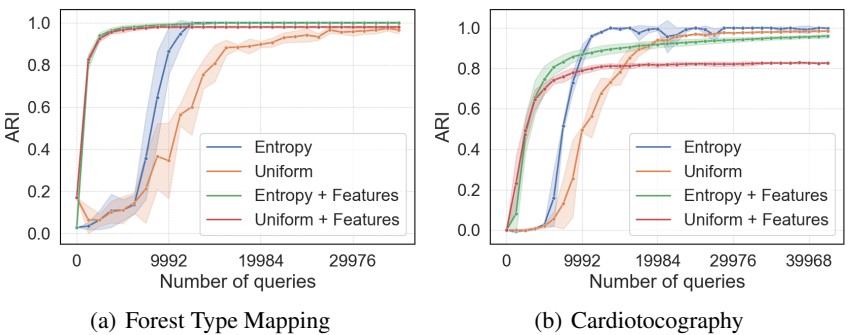

(a) Forest Type Mapping          (b) Cardiotocography

Figure 16: Performance of $a^{\text{Entropy}}$ and $a^{\text{Uniform}}$ when leveraging feature vectors for two real-world datasets.

Then, we train a pairwise similarity prediction model $f_\theta : \mathbf{X} \times \mathbf{X} \to [-1, +1]$ using the collected dataset. The training process follows a procedure similar to Oracle 4 (see Appendix C.2 for details). Finally, use $f_\theta$ to predict similarities for the pairs that have not yet been queried. To minimize noise or incorrect predictions, it may be advantageous to limit predictions to pairs for which $f_\theta$ exhibits high confidence.

We conducted some experiments to evaluate this approach, with the results presented in Figures 15-16. The experiments in Figure 15 were performed on two synthetic datasets. The first dataset has relatively simple structure, with well-separated clusters, while the second dataset is more challenging, exhibiting overlap between clusters. The results suggest that the prediction component facilitates rapid convergence, reducing the number of queries required. However, we observe that leveraging

features can lead to convergence toward arbitrarily poor solutions if the dataset contains bias. This issue is particularly pronounced in the more challenging dataset, where noise and bias are more prevalent. Figure 16 presents the results for two real-world datasets, showcasing a similar trend. Notably, in the forest type mapping dataset, the ground-truth clustering is identified rapidly when feature vectors are used. In contrast, for the cardiotocography dataset, the procedure converges to a suboptimal clustering when using features, likely due to noise or bias in the dataset's feature space.

It is important to emphasize that the primary focus of this paper is the setting where high-quality feature vectors are not assumed to be available. Therefore, the proposed prediction component should be viewed as an intriguing direction for future work in this context. However, since we do not assume access to feature vectors, it would not be appropriate to center this study on the predictive component (or any other way of incorporating feature vectors).

## D  MAX CORRELATION CLUSTERING ALGORITHM

In this section, we describe the CC algorithm used in the active CC procedure outlined in Section 2.2. The algorithm was derived in (Aronsson & Chehreghani, 2024) based on the max correlation cost function $R^{\mathrm{MC}}(c \mid S) \triangleq - \sum_{\substack{(u,v) \in \mathcal{E} \\ c_u = c_v}} S_{uv}$ introduced in Proposition 2.1. It is highly robust to noise in $S$ and dynamically determines the number of clusters.

The method is based on local search and is outlined in Alg. 5. It takes as input a set of objects $\mathcal{V}$, a similarity matrix $S$, an initial number of clusters $K$, the number of repetitions $T$, and a stopping threshold $\eta$. In our experiments, we set $T = 5$, $\eta = 2^{-52}$ (double precision machine epsilon) and $K = |\mathcal{V}|$ in the first iteration of the active CC procedure, and then $K = K^i$ for all remaining iterations where $K^i$ denotes the number of clusters in the current clustering $c^i$. The output is a clustering $c \in \mathcal{C}$. The main part of the algorithm (lines 4-22) is based on the local search of the respective non-convex objective. Therefore, we run the algorithm $T$ times with different random initializations and return the best clustering in terms of the objective function. The main algorithm (starting from line 4) consists of initializing the current clustering $c$ randomly. Then, it loops for as long as the current *max correlation* objective changes by at least $\eta$ compared to the last iteration. If not, we assume it has converged to some (local) optimum. Each repetition consists of iterating over all the objects in $\mathcal{V}$ in a random order $\mathcal{V}_{\mathrm{rand}}$ (this ensures variability between the $T$ runs). For each object $u \in \mathcal{V}_{\mathrm{rand}}$, it calculates the similarity (correlation) between $u$ and all clusters $k \in \{1, \ldots, K\}$, which is denoted by $S_k(u)$. Then, the cluster $k_{\mathrm{max}}$ that is most similar to $u$ is obtained. Now, if the most similar cluster to $u$ has a negative correlation score, this indicates that $u$ is not sufficiently similar to any of the existing clusters. Thus, we construct a new cluster with $u$ as the only member. If the most similar cluster to $u$ is positive, we simply assign $u$ to this cluster. Consequently, the number of clusters will dynamically change based on the pairwise similarities (it is possible that the only object of a singleton cluster is assigned to another cluster and thus the singleton cluster disappears). Finally, in each repetition the current max correlation objective is computed efficiently by only updating it based on the current change of the clustering $c$ (i.e., lines 14 and 20). The computational complexity of the procedure is $O(KN^2)$. See (Aronsson & Chehreghani, 2024) for additional details about how the algorithm was derived.

## E  RELATION TO MULTI-ARMED BANDIT METHODS

The studies in (Gullo et al., 2023; Kuroki et al., 2024) address query-efficient correlation clustering (CC) by framing it as a *multi-armed bandit* (MAB) problem. Below, we compare these approaches to our own methods.

First, (Gullo et al., 2023) distinguish their approach from the query-efficient CC framework studied by (Bressan et al., 2019; García-Soriano et al., 2020), which is the setting we also consider. The key difference lies in the assumption regarding the query budget $B$. While we assume a fixed budget with $B \ll |\mathcal{E}|$, (Gullo et al., 2023) does not impose this constraint, allowing the number of queries to exceed the total number of pairs.

In (Kuroki et al., 2024), the authors propose learning edge weights (pairwise similarities) using combinatorial bandit algorithms. Similar to other query-efficient CC studies (Bressan et al., 2019; García-Soriano et al., 2020), they utilize KwikCluster as their base clustering algorithm. KwikCluster,

---

**Algorithm 5** Max Correlation Clustering Algorithm $\mathcal{A}$ (dynamic $K$)

---

**Input**: $\mathcal{V}$, $\boldsymbol{S}$, initial number of clusters $K$, number of iterations $T$, stopping threshold $\eta$
**Output**: Clustering solution $\boldsymbol{c} \in \mathcal{C}$

1: $N \leftarrow |\mathcal{V}|$
2: $R_{\text{best}}^{\text{MC}} \leftarrow -\infty$
3: **for** $j \in \{1, \ldots, T\}$ **do**
4:     $\boldsymbol{c} \leftarrow$ random clustering in $\mathcal{C}$ with $K$ clusters
5:     $R^{\text{MC}} \leftarrow R^{\text{MC}}(\boldsymbol{c} \mid \boldsymbol{S})$
6:     $R_{\text{old}}^{\text{MC}} \leftarrow R^{\text{MC}} - 1$
7:     **while** $|R^{\text{MC}} - R_{old}^{\text{MC}}| > \eta$ **do**
8:         $R_{old}^{\text{MC}} \leftarrow R^{\text{MC}}$
9:         $\mathcal{V}_{\text{rand}} \leftarrow$ a random permutation of the objects in $\mathcal{V}$
10:        **for** each $u$ in $\mathcal{V}_{\text{rand}}$ **do**
11:            $S_k(u) \leftarrow \sum_{v:c_v^i = k} S_{uv}, \quad \forall k \in \{1, \ldots, K\}$
12:            $k_{\max} \leftarrow \arg\max_{k \in \{1, \ldots, K\}} S_k(u)$
13:            **if** $S_{k_{\max}}(u) < 0$ **then**
14:                $R^{\text{MC}} \leftarrow R^{\text{MC}} - S_{c_u}(u)$
15:                $c_u \leftarrow K + 1$
16:                $K \leftarrow K + 1$
17:            **else**
18:                $k_{\text{old}} \leftarrow c_u$
19:                $c_u \leftarrow k_{\max}$
20:                $R^{\text{MC}} \leftarrow R^{\text{MC}} - S_{c_u}(u) + S_{k_{\max}}(u)$
21:                If cluster $k_{\text{old}}$ is now empty, decrement $c_v$ for all $v \in \mathcal{V}$ for which $c_v > k_{\text{old}}$, and then decrement $K$.
22:            **end if**
23:        **end for**
24:    **end while**
25:    **if** $R^{\text{MC}} > R_{\text{best}}^{\text{MC}}$ **then**
26:        $\boldsymbol{c}_{\text{best}} \leftarrow \boldsymbol{c}$
27:        $R_{\text{best}}^{\text{MC}} \leftarrow R^{\text{MC}}$
28:    **end if**
29: **end for**
30: **return** $\boldsymbol{c}_{\text{best}}$

---

being a pivot-based algorithm, is particularly sensitive to noise. Their approach maps each edge to a distinct arm (resulting in $O(N^2)$ arms), and they apply combinatorial bandit algorithms to estimate similarities. However, this approach presents several limitations:

- The algorithms are tailored to satisfy specific theoretical properties, which impose practical constraints (discussed below).

- They assume non-persistent noise, meaning multiple queries of the same similarity (or multiple pulls of the same arm) are permitted—this is a standard assumption in the MAB literature. In contrast, our methods are robust even under persistent noise, where only a single query per pair is allowed.

- Their strategy for selecting which similarities to query is limited, as it does not account for correlations between pairs (arms), unlike our approach, which incorporates model uncertainty to guide query selection.

- They consider two primary settings:
    - Fixed Confidence (KF-FC): This setting requires each arm to be pulled at least once, leading to more queries than the total number of pairwise similarities—this is a common assumption for many MAB algorithms. Our methods, on the other hand, achieve effective clustering with significantly fewer queries. Additionally, their framework does not accommodate a predefined query budget $B$.
    - Fixed Budget (KF-FB): Although this setting allows for a predefined budget $B$, the smallest budget considered in (Kuroki et al., 2024) is $N^{2.1}$, which exceeds the total

number of pairs. This constraint arises from the requirements of Algorithm 3 in (Kuroki et al., 2024), which necessitates a budget larger than $N^2$ to function properly, ensuring the validity of their theoretical analysis. As demonstrated via extensive experiments, our methods require significantly fewer queries.

