# OpenReview forum: "Information-Theoretic Active Correlation Clustering"
_ICLR.cc/2025/Conference — Submitted to ICLR 2025_

### Official Review · Reviewer_Yp7c · 2024-10-29

**Soundness:** 3
**Presentation:** 3
**Contribution:** 2
**Rating:** 6
**Confidence:** 4

**Summary:**

This paper proposes an active correlation clustering method based on information theory. It applies the mean-field approximation to approximate the Gibbs distribution and proposes four acquisition functions for active correlation clustering based on the approximation.

**Strengths:**

Strengths:
1. The idea of information-theoretic active correlation clustering together with the mean-field approximation is interesting.
2. The experimental results are good.
3. The presentation and organization are good.

**Weaknesses:**

Weaknesses and Questions:
1. The setting where we can only access to noisy similarities of some pairs is interesting. However, it would be better to introduce some application scenarios in real world, and apply at least one of the real-world data sets in the experiments instead of synthetically adding some noises to the ground truth, because the oracles used in the experiments may not follow the distribution of real-world noises.

2. The mean-field approximation method seems similar to the widely-used variational inference. What are the differences between the mean-field approximation and variational inference?

3. In many data sets, the paper only uses a subset. Why not use the whole data set? Cannot the proposed method handle large-scale data sets?

4. In some data sets of Figures 1 and 2, the results seem incomplete. For example, in Figure 1(g) and 1(h), the results of EIG-O and EIG-P are missing. Why?

**Questions:**

Please See Weaknesses.

---

> ### Author Response · Authors · 2024-11-17
> **Response to Reviewer Yp7c**
>
> Thank you for the review and the valuable comments/suggestions! We have attempted to address them in the following.
>
> * **The setting where we can only access to noisy similarities of some pairs is interesting. However, it would be better to introduce some application scenarios in real world, and apply at least one of the real-world data sets in the experiments instead of synthetically adding some noises to the ground truth, because the oracles used in the experiments may not follow the distribution of real-world noises.**
>
> In our experimental studies, we examine four different oracles, which can be grouped into two types: (1) those that provide unbiased similarities with synthetically added noise (Oracles 1–3) and (2) those that generate similarities directly from the feature space of a given real-world dataset without any additional synthetic noise (Oracle 4). Thus, all experiments involving Oracle 4 address the exact scenario described by the reviewer. For example, see Figure 2 in the main paper.
>
> * **The mean-field approximation method seems similar to the widely-used variational inference. What are the differences between the mean-field approximation and variational inference?**
>
> In variational inference, the objective is to approximate an intractable distribution $P$ with a more tractable distribution $Q$. When $Q$ is assumed to factorize as $Q(x) = \prod_i Q(x_i)$, this approach is referred to as a mean-field approximation [1]. In essence, the mean-field approximation represents a special case of variational inference where $Q$ is assumed to be a factorial distribution.
>
> The iterative EM-style procedure derived in Theorem 3.1 optimizes the Evidence Lower Bound (ELBO), as demonstrated in the proof of Theorem 3.1, aligning it with the standard framework of variational inference [1]. However, our setting diverges from traditional variational inference in two key ways: (1) it is non-parametric, as the intractable Gibbs distribution in Eq. 3 is entirely defined by the data (i.e., the pairwise similarities), and (2) it operates in a non-Bayesian context.
>
> * **In many data sets, the paper only uses a subset. Why not use the whole data set? Cannot the proposed method handle large-scale data sets?**
>
> The acquisition functions EIG-O and EIG-P are currently constrained to smaller problem instances, as discussed in lines 309–314 and further elaborated in Appendix C.6, where we analyze the runtime of all acquisition functions. This limitation is addressed in several ways:
>
> 1. Algorithm 3: We propose Algorithm 3, which partially alleviates the scalability issues of EIG-O and EIG-P, although not entirely (see lines 309–314 for further details).
>
> 2. JEIG Acquisition Function: We introduce JEIG as an alternative approximation for the information gain. JEIG is competitive with EIG-O and EIG-P in terms of performance, but at the same time, significantly more scalable.
>
> The primary reason we limit the size of all datasets to at most $N = 1,000$ is the inefficiency of EIG-O and EIG-P for larger datasets. All other methods, including Entropy and JEIG, can scale effectively to larger problem sizes.
>
> Finally, we believe that similar to JEIG, there are other approximations for EIG-O and EIG-P that could significantly improve their scalability. Exploring such approximations presents an interesting avenue for future research in this context.
>
> * **In some data sets of Figures 1 and 2, the results seem incomplete. For example, in Figure 1(g) and 1(h), the results of EIG-O and EIG-P are missing. Why?**
>
> The reason has been explained in lines 482–485 of the paper. As mentioned earlier, EIG-O and EIG-P are generally consistent with the performance of JEIG because they approximate the same quantity (though in different ways). However, JEIG is significantly computationally more efficient. This is consistently evident from results such as Figure 1 and some plots in Figure 2. Due to their computational inefficiency, we occasionally excluded EIG-O and EIG-P from certain experiments, as JEIG serves as a reliable and efficient representation of their performance.
>
> **References**
>
> [1] David M. Blei, Alp Kucukelbir, Jon D. McAuliffe: Variational Inference: A Review for Statisticians. CoRR abs/1601.00670 (2016).

---

> > ### Comment · Reviewer_Yp7c · 2024-11-26
> >
> > Thanks for your responses, which addressed some of my concerns. I'd like to keep my score.

---

### Official Review · Reviewer_Kxxe · 2024-10-30

**Soundness:** 3
**Presentation:** 3
**Contribution:** 2
**Rating:** 8
**Confidence:** 4

**Summary:**

This manuscript presents information-theoretic acquisition functions for active correlation clustering, where pairwise similarities are queried from an oracle to construct clusters. The authors introduce four acquisition functions: Entropy, EIG-O, EIG-P, and JEIG. mean-field approximations are adopted to make computations tractable. Through extensive experiments across multiple datasets and noisy oracles of various types, the proposed approaches show superior performance compared to existing methods.

**Strengths:**

The authors proposed novel information-theoretic acquisition functions to active correlation clustering. Mean-field approximations are derived for these functions.

The paper presents comprehensive empirical validation across diverse datasets. The authors discuss how noisy oracles with different types impact the performance of the model.

The authors utilize clear mathematical notation and describe the algorithms in detail. The comprehensive appendices provide details in the implementation and in the experiments.

**Weaknesses:**

This manuscript proposes a correlation clustering method that operates solely on pairwise similarities, without utilizing sample features. A significant limitation is the method's extensive labeling requirements. As shown in the experiments, for datasets with no more than 1,000 samples, thousands of pairwise labels are needed to achieve acceptable performance. This high labeling burden makes the method impractical for real-world applications, where collecting pairwise labels is typically more resource-intensive than collecting instance-level labels. Given these practical constraints, the paper's impact is likely to be limited despite its contributions.

**Questions:**

How does the cluster imbalance impact the performance of the proposed method?

---

> ### Author Response · Authors · 2024-11-17
> **Response to Reviewer Kxxe**
>
> Thank you for the review and the valuable comments/suggestions! We have attempted to address them in the following.
>
> * **A significant limitation is the method's extensive labeling requirements. As shown in the experiments, for datasets with no more than 1,000 samples, thousands of pairwise labels are needed to achieve acceptable performance. This high labeling burden makes the method impractical for real-world applications, where collecting pairwise labels is typically more resource-intensive than collecting instance-level labels. Given these practical constraints, the paper's impact is likely to be limited despite its contributions.**
>
> We argue that we obtain promising results after only a small portion of all possible queries is made. For instance, in Figure 2, we observe that the procedure converges after approximately $8,500$ queries when using the JEIG acquisition function. With $N=1,000$ samples, the total number of pairwise similarities is $\frac{N \times (N-1)}{2} = 499,500$. This means our active correlation clustering procedure requires only about 1.7% of the total pairwise similarities to converge.
>
> All previous works, which we categorize under the name “active correlation clustering”, also require many queries, but still very small in relation to total number of pairwise similarities as illustrated above. See lines 60-77 for a discussion about these works. The primary reason for the large number of queries in this setting is because access to feature vectors is not assumed. This means that we rely solely on the information coming from the oracle in the form of pairwise similarities (see motivation for this in next paragraph). This combined with the use of a noisy oracle naturally requires more queries to the oracle.
>
> Clustering is inherently an underspecified problem, often leading to arbitrary clustering solutions that fail to align with user expectations [1]. This challenge arises from noise or ambiguity in the feature space, which can bias the resulting solutions. Active correlation clustering offers a systematic approach to mitigate this bias by selectively incorporating information about a subset of pairwise similarities. This targeted approach ensures that the selected similarities more accurately reflect the true underlying clustering structure. In particular, if the oracle can provide similarities which are entirely unbiased (but possibly noisy), we can effectively recover the ground-truth clustering. This is illustrated in our experiments for oracles 1-3. In practical applications, similarities may be generated in a semi-automated manner—for example, derived from a predictive model with no or partial human supervision, as discussed in [2]. This semi-automated approach ensures the feasibility of active correlation clustering, even in scenarios where a large number of queries are required.
>
> Finally, we respectfully disagree with the reviewers’ argument that “collecting pairwise labels is typically more resource-intensive than collecting instance-level labels.” Evidence from various areas of machine learning suggest that providing pairwise feedback is generally more intuitive and less demanding than assigning instance-level labels. For example, consider a scenario where a human annotator is presented with two images. It is often easier for the annotator to determine whether the images are similar or dissimilar than to assign a specific class to each image. The latter task typically requires a deeper understanding of domain-specific knowledge about the classes involved.
>
> A concrete example of this principle can be seen in Reinforcement Learning from Human Feedback (RLHF), a method widely used to fine-tune large language models. RLHF leverages pairwise preferences because it is much simpler for humans to indicate which of two candidate responses to a prompt is better than to independently identify or construct an ideal response [3]. This highlights the efficiency and accessibility of pairwise feedback compared to instance-level labeling.
>
> * **How does the cluster imbalance impact the performance of the proposed method?**
>
> Many of the datasets considered consist of highly imbalanced clusters (for example, the Ecoli dataset). See page 21 for information about the class balance of all datasets. Despite this cluster imbalance, our methods still perform well for these datasets.
>
> **References**
>
> [1] Sugato Basu et all., Active Semi-Supervision for Pairwise Constrained Clustering. SDM 2004.
>
> [2] Sandeep Silwal et al. KwikBucks: Correlation Clustering with Cheap-Weak and Expensive-Strong Signals. ICLR 2023,
>
> [3] Long Ouyang et al. Training language models to follow instructions with human feedback. NeurIPS 2022.

---

> ### Author Response · Authors · 2024-11-19
> **Response to reviewer Kxxe**
>
> Thank you for the response. We will split our response into three posts, with references added to the last.
>
> * **In contrast, feature-based active learning approaches that collects instance labels typically require only tens of queries to achieve comparable clustering performance.**
>
> Please note that in this work, we study (active) *correlation clustering*. One of the main motivations for correlation clustering (whether it is active or not) is that *access to feature vectors is not assumed* [3, 8]. In other words, correlation clustering provides a way to generate high-quality clustering solutions in cases where high-quality feature vectors are not assumed to be available (since we only need similarities) [3]. “Active correlation clustering”, also called “query-efficient correlation clustering”, extends this by addressing cases where computing all $O(N^2)$ similarities is impractical or costly.
>
> *Our setting is fundamentally different from "feature-based active learning”, as correlation clustering assumes no access to feature vectors* [3, 8]. In practice, feature vectors may be unavailable due to high acquisition costs (see the 'Active Feature Acquisition' problem). Alternatively, feature vectors might be available but of very low quality (i.e., highly biased, leading to poor clustering solutions).  Prior work on active correlation clustering, including recent studies [4, 5, 6] published at ICLR and NeurIPS, also requires many queries for the same reasons. In fact, they generally require more queries than us, see Figure 1 of [1] for example (we use the same framework as [1], with significantly improved acquisition functions). Importantly, in correlation clustering, similarities are not always provided by a human oracle [3]; they could be generated by automated prediction models (e.g., large deep learning models) that produce high-quality similarities but incur significant computational costs for each inference call. Humans may or may not be involved in this process (see [4] for further discussion). For example, making $499,500$ inference calls might be infeasible, while $8,500$ calls could still be practical (based on the example in our previous response).
>
> The reviewer suggests that feature-based methods collecting instance labels achieve comparable clustering performance. This is a fairly broad generalization, and will not always be true. As noted, exploiting information from a biased feature space often leads to worse performance. While a high-quality feature space with clear/separable clusters would yield better results, such scenarios eliminate the need for constrained clustering, as the ground-truth clustering is already evident. Thus, while the methods the reviewer references might require fewer queries to "converge”, their solutions can be significantly worse due to feature space noise or bias. This is demonstrated concretely in [1], which compares active correlation clustering to the feature-based active pairwise constraint clustering approach COBRAS [2] (e.g., see Figure 1 of [1]).
>
> (continued)

---

> ### Author Response · Authors · 2024-11-19
> **Response to reviewer Kxxe**
>
> * **The Figure 1 in the paper of Hongfu et al. (2015) demonstrates that pairwise similarity labels are often more ambiguous and challenging to obtain than direct labels**
>
> Figure 1 of Hongfu et al. (2015) argue that pairwise feedback may generally be more *noisy* than instance-level feedback but does not claim that pairwise queries are more *expensive* to obtain. However, this does not mean pairwise feedback is always more noisy in all practical settings. Hongfu et al. (2015) are simply arguing that this may occur potentially, and propose a possible solution. They highlight that existing active constraint clustering methods perform poorly due to their sensitivity to noise in pairwise constraints. In contrast, a key property of our framework is its robustness to such noise and inconsistencies, as demonstrated extensively in our experiments. This robustness is further supported in [1], which compares active correlation clustering with the feature-based active constraint clustering method COBRAS [2].
>
> More broadly, this robustness is an inherent feature of correlation clustering. In correlation clustering, each similarity acts as a "soft constraint”. For example, a positive similarity of +1 between objects $u$ and $v$ does not strictly enforce their placement in the same cluster. Instead, the goal is to minimize the correlation clustering cost (Eq. 1), resulting in a solution that is as consistent as possible with the provided similarities. This flexibility naturally leads to greater resilience against noise and inconsistencies compared to active constraint clustering methods. In fact, using correlation clustering to “denoise” constraints for existing active pairwise constraint clustering methods, is an interesting avenue for future research.
>
> While we respect the opinion of the reviewer, it seems highly unfair to discredit a large body of work on active clustering approaches that deal with pairwise constraints/similarities (see [7] for an overview of such methods) based on a single paper that happens to argue for instance-level constraints. In addition, this paper is from 2015 and a lot of progress has been made since then in terms of improving the usefulness of pairwise constraints/similarities for clustering (as discussed above).
>
> (continued)

---

### Official Review · Reviewer_B3JR · 2024-11-01

**Soundness:** 2
**Presentation:** 2
**Contribution:** 2
**Rating:** 3
**Confidence:** 2

**Summary:**

The paper addresses an interesting problem: mapping data points into distinct clusters with high accuracy by querying the similarity between them a limited number of times. The objective function, as outlined in Equations (1) and (2), aims to minimize clustering disagreement. However, the rationale behind the objective function is somewhat unclear, as it appears that clustering disagreement may also be influenced by the scale of the data points.

The main theoretical guarantee presented is a local minimum in Theorem 3.1, though it is unclear how this guarantee compares to those in existing literature. Additionally, it would be helpful to understand the authors' considerations regarding the number of clusters, as how to infer the number of clusters from the structure of the similarity matrix graph.

Notably, this approach diverges from the usual practice of providing a guarantee for the distance between ground truth and predicted cluster centers. Achieving the guarantee in Theorem 3.1 involves multiple steps of relaxation by introducing concepts such as information gain, entropy, and mean-field approximation, which raises questions about the accuracy of the theorem.

The technical details in Algorithms 1 and 2 are not discussed. Algorithm 1 (on page 4) queries similarity through an oracle, producing a similarity matrix with both positive and negative values. A positive value indicates that a data pair belongs to the same class, while a negative one implies otherwise. This differs from the standard approach of thresholding positive similarity values, so I am curious about how this algorithm would perform in real-world applications. Additionally, the "budget," a critical parameter, is not discussed. In Algorithm 2 (on page 5), the criteria for defining the convergence of Q in Step 2 remain unclear.

The experimental results are a positive aspect of a theoretical paper.

**Strengths:**

The paper addresses an interesting problem: mapping data points into distinct clusters with high accuracy by querying the similarity between them a limited number of times.

The experimental results are a positive aspect of a theoretical paper.

**Weaknesses:**

It is not clear if the objective function makes sense. The objective function, as outlined in Equations (1) and (2), aims to minimize clustering disagreement. However, the rationale behind the objective function is somewhat unclear, as it appears that clustering disagreement may also be influenced by the scale of the data points.


The technical details in Algorithms 1 and 2 are not discussed. For example, the "budget" in Algorithm 1 (on page 4)  a critical parameter, is not discussed.  In Algorithm 2 (on page 5), the criteria for defining the convergence of Q in Step 2 remain unclear.

**Questions:**

The objective function, as outlined in Equations (1) and (2), aims to minimize clustering disagreement. However, the rationale behind the objective function is somewhat unclear, as it appears that clustering disagreement may also be influenced by the scale of the data points.

The main theoretical guarantee presented is a local minimum in Theorem 3.1, though it is unclear how this guarantee compares to those in existing literature.

It would be helpful to understand the authors' considerations regarding the number of clusters, as how to infer the number of clusters from the structure of the similarity matrix graph.

How do authors define "budget" in Algorithm 1? In Algorithm 2 (on page 5), the criteria for defining the convergence of Q in Step 2 remain unclear.

---

> ### Author Response · Authors · 2024-11-17
> **Response to Reviewer B3JR**
>
> Thank you for the review and useful comments! We have attempted to address them in the following. The rebuttal is split into two responses, with references added to the last
>
> * **The technical details in Algorithms 1 and 2 are not discussed. Algorithm 1 (on page 4) queries similarity through an oracle, producing a similarity matrix with both positive and negative values. A positive value indicates that a data pair belongs to the same class, while a negative one implies otherwise. This differs from the standard approach of thresholding positive similarity values, so I am curious about how this algorithm would perform in real-world applications.**
>
> In correlation clustering, the use of both positive and negative similarities provides a more expressive framework for capturing the relationship between data points. Positive similarities indicate that two data points are likely to belong to the same cluster, while negative similarities suggest they belong to different clusters. This approach aligns with the intrinsic goal of correlation clustering, which seeks to partition data such that the total agreement with given similarities is maximized.
>
> By incorporating negative similarities, the algorithm can leverage both agreements and disagreements between data pairs, which is particularly beneficial in scenarios where dissimilarity information is as crucial as similarity information. This differs from other approaches that consider only positive similarities and ignore the potential value of explicit negative relationships.
>
> In real-world applications, this can enhance performance in domains with clear notions of both similarity and dissimilarity, such as social networks, document clustering, or recommendation systems, where negative interactions or disagreements provide critical context. Using both positive and negative similarities allows the algorithm to represent and process a richer set of constraints, leading to more accurate and meaningful clustering solutions in practice.
>
> Consider the following concrete example (from [3], also cited in our paper). In a social network, friendships can be represented by positive similarities, while dislike or antagonistic relationships are captured by negative similarities. Incorporating both allows the algorithm to distinguish cohesive communities from opposing groups, which would be missed by considering only positive similarities. This richer representation leads to more meaningful clustering in such contexts.
>
> * **The objective function, as outlined in Equations (1) and (2), aims to minimize clustering disagreement. However, the rationale behind the objective function is somewhat unclear, as it appears that clustering disagreement may also be influenced by the scale of the data points.**
>
> Equation 1 represents the standard formulation of correlation clustering, where similarities can take any positive or negative value [1]. This formulation is not introduced by us in this paper; it is a well-established objective for correlation clustering. Minimizing this objective yields the clustering solution that best aligns with the provided similarities, effectively maximizing intra-cluster similarities while minimizing inter-cluster similarities.
>
> Moreover, the scale of the similarities influences the minimization of Eq. 1. However, this highlights the inherent flexibility of correlation clustering, which can accommodate a wide range of similarity values. For instance, assigning higher absolute similarity values to certain pairs can prioritize them over other pairs. As an example, assume we know with full certainty that objects $u$ and $v$ should be placed in the same cluster. Then, by setting the similarity $S_{uv}$ to a large value, we can enforce this. However, potential concerns about the scale of similarities can be easily mitigated through appropriate preprocessing/normalization steps.
>
> * **The main theoretical guarantee presented is a local minimum in Theorem 3.1, though it is unclear how this guarantee compares to those in existing literature.**
>
> Theorem 3.1 yields a specialized EM-style algorithm that computes a local minimum of the KL-divergence (Eq. 4) specialized for correlation clustering (minimizing the function is an NP-hard problem and thus computing a local minimum is already a good step). This is consistent with work in [2] demonstrating that an EM-style procedure yields minimizing the KL-divergence for a broader class of clustering cost functions, though investigated for clustering models that deal with only non-negative pairwise similarities. The way we specialize this to our setting (correlation clustering) results in an efficient way of performing the E and M steps utilizing the vectorized forms. Please see lines 216-235 for more details.

---

> ### Author Response · Authors · 2024-11-17
> **Response to Reviewer B3JR**
>
> * **It would be helpful to understand the authors' considerations regarding the number of clusters, as how to infer the number of clusters from the structure of the similarity matrix graph.**
>
> Our procedure automatically determines the number of clusters, meaning it does not need to be specified in advance. This is achieved through the correlation clustering algorithm utilized in line 4 of Algorithm 1. As detailed in Appendix D (specifically Algorithm 5), this algorithm dynamically identifies the optimal number of clusters based on the current pairwise similarities. This is a unique advantage of correlation clustering compared to many other clustering models.
>
> * **How do authors define "budget" in Algorithm 1? In Algorithm 2 (on page 5), the criteria for defining the convergence of Q in Step 2 remain unclear.**
>
> In active learning, the budget refers to the number of queries allowed before the procedure is stopped. Importantly, our method will always produce a clustering solution, regardless of the budget. However, as expected, the quality of the solution improves with larger budgets.
>
> In practical scenarios, the budget often reflects resource constraints for obtaining similarities. For instance, if the oracle is a human annotator (compensated on an hourly basis), financial constraints may limit the number of similarities that can be acquired. In contrast, in a controlled experimental setting, all similarities are accessible, allowing us to evaluate performance across different budget levels. This enables a comprehensive analysis of how varying the budget impacts clustering quality, as demonstrated in our experiments.
>
> Convergence of $Q$ in Algorithm 2 is determined by evaluating whether the difference between $Q^{t-1}$ and $Q^t$ is sufficiently small. Specifically, this can be assessed by checking if the Frobenius norm of their difference falls below a predefined threshold $\eta$.
>
> **References**
>
> [1] Francesco Bonchi, David García-Soriano, Edo Liberty: Correlation clustering: from theory to practice. KDD 2014.
>
> [2] Thomas Hofmann, Jan Puzicha, Joachim M. Buhmann: Unsupervised Texture Segmentation in a Deterministic Annealing Framework. IEEE Trans. Pattern Anal. Mach. Intell. 20(8): 803-818 (1998).
>
> [3] Jiliang Tang et al. A Survey of Signed Network Mining in Social Media. ACM Comput. Surv. 49(3): 42:1-42:37 (2016)

---

> > ### Comment · Reviewer_B3JR · 2024-11-18
> >
> > I have reviewed the authors' response and appreciate that they have addressed some of my concerns. I understand that the primary focus of this work is on proposing a practical solution. Based on this, I would like to raise my rating to 5.

---

> ### Author Response · Authors · 2024-11-20
> **Response to reviewer B3JR**
>
> We appreciate your feedback on our response.
>
> Please note that in the revised version uploaded after your latest comment, we have added Section C.8 (to the Appendix), where we investigate how our framework can incorporate features in a generic way. We also demonstrate the impact of feature quality on the effectiveness of active learning.
>
> We have done our best to address all of your questions and concerns. If there is any ambiguity or if you feel that any questions have been left unaddressed, we kindly ask you to remind us.
>
> Finally, we would like to kindly remind the reviewer that the score has not been updated to 5. This can be updated by editing the original review.

---

### Official Review · Reviewer_gKTJ · 2024-11-02

**Soundness:** 4
**Presentation:** 3
**Contribution:** 2
**Rating:** 6
**Confidence:** 3

**Summary:**

The paper presents a new approach to correlation clustering with a limited budget for similarity queries between pair objects. The main contribution is the design of new techniques (the so-called acquisition functions) for selecting object pairs whose similarity should be computed by a noisy oracle and which can then be used as input to a correlation clustering algorithm.

**Strengths:**

- The considered problem is important and has practical applications.
- An interesting approach for the selection of object pairs whose similarity should be computed.
- Good experimental results.

**Weaknesses:**

- The approach is just a heuristic, there are no approximation guarantees, not even an attempt to discuss the quality of approximation from a theoretic point of view.
- The writing can be improved. For example, Theorem 1 is not central to the presented approach, it is rather about an efficient approximation of Gibbs sampling. I think the presentation should first focus on the 4 acquisition functions and then explain how to approximate Gibbs sampling.
- I couldn't understand why only noisy oracles are considered, see my question below.

**Questions:**

I don't understand why only noisy oracles are considered. I would like to know how the approach performs if the oracle returns the precise pair similarity. This is a realistic and practical use case and I think it should be considered. Would we be then able to obtain some approximation guarantees, maybe after assuming some properties of the acquisition function?

---

> ### Author Response · Authors · 2024-11-17
> **Response to Reviewer gKTJ**
>
> Thank you for the review and the valuable comments! We have attempted to address them in the following. The rebuttal is split into two responses, with references added to the last
>
> * **The approach is just a heuristic, there are no approximation guarantees, not even an attempt to discuss the quality of approximation from a theoretic point of view.**
>
> In active learning, research typically follows two main directions: theoretical and practical. For instance, in "standard" active learning, where the goal is to query the labels of data points [1], theoretical studies such as [2] focus on designing algorithms with upper bounds on query complexity. However, these studies often suffer from significant limitations: (1) they rely on unrealistic assumptions, such as assuming zero noise in the data (e.g., perfectly separable classes in classification problems), or (2) they propose algorithms that are computationally impractical for real-world applications. On the other hand, practical studies, such as [3], aim to develop active learning methods tailored to realistic, complex scenarios. While these approaches are more applicable in practice, they typically do not provide theoretical approximation guarantees due to the inherent complexity of the settings they address.
>
> In the context of active learning for correlation clustering (i.e., the focus of our work), the majority of research to date has leaned heavily toward theoretical approaches (as discussed in detail in lines 68–77). Notably, the QECC algorithm, introduced by [4], is an example of this. However, as in standard active learning settings, these approaches often rely on restrictive assumptions. Specifically, their theoretical analysis is constrained to the well-known pivot-based correlation clustering algorithm, KwikCluster, which is highly sensitive to noise and therefore unsuitable for many practical applications.
>
> In contrast, the work by [5] was the first to explore active correlation clustering with an emphasis on practicality. Their framework allows for the use of any correlation clustering algorithm, enabling practitioners to select the most appropriate algorithm for a given problem (see Algorithm 1 of our paper). Our work builds upon [5], continuing the effort to develop methods that are applicable to real-world scenarios and are more effective than those in [5]. This focus on practicality inherently introduces additional challenges for theoretical analysis.  Due to the aforementioned complexities, information-theoretic active learning methods even in standard settings usually do not lead to theoretical guarantees.
>
> It may be possible to establish approximation guarantees for Algorithm 1 by imposing several restrictive assumptions regarding (1) the acquisition function, (2) the correlation clustering algorithm, and (3) the oracle. However, our primary focus is on advancing active correlation clustering toward practical applicability. We consider our work a significant step in this direction and leave the exploration of such theoretical analyses for future research.
>
> * **The writing can be improved. For example, Theorem 1 is not central to the presented approach, it is rather about an efficient approximation of Gibbs sampling. I think the presentation should first focus on the 4 acquisition functions and then explain how to approximate Gibbs sampling.**
>
> Thank you for the suggestion. We did consider presenting the paper in this manner, but ultimately chose the current structure for the following reason: the derivation of Alg. 2 (mean-field approximation), which builds on Theorem 3.1, is a central component of the paper, as all acquisition functions heavily rely on it. That said, we greatly appreciate the reviewer’s suggestion and acknowledge it fully.

---

> ### Author Response · Authors · 2024-11-17
> **Response to Reviewer gKTJ**
>
> * **Only noisy oracles are considered (can we derive theory if no noise assumed?)**
>
> In real-world scenarios, pairwise similarities often contain some level of noise, as for example the unknown underlying feature space from which they are derived typically exhibits inherent noise or ambiguity. This is reflected in our experiments with Oracle 4, where the only source of noise in the resulting similarities originates from the feature space itself—no additional artificial noise is introduced. On the other hand, the annotator/oracle is simply uncertain about the correct label/feedback, as reflected in our other oracle models.
>
> Figure 3a illustrates the performance of our methods across varying noise levels for Oracle 1, including a zero-noise scenario. Our proposed acquisition functions demonstrate the best performance even in the absence of noise, though the improvement over baselines is smaller (albeit still significant). This aligns with prior work on active learning, where the advantages of acquisition functions over random selection tend to increase in more complex—and therefore more realistic—settings.
>
> Furthermore, as noted in the footnote on page 8, assuming zero noise for Oracles 1–3 simplifies the correlation clustering problem (i.e., minimizing Eq. 1), making it no longer NP-hard. In such cases, the problem becomes trivial and less relevant for study.
>
> Finally, assuming zero noise is likely insufficient for deriving theoretical guarantees in this setting. Additional assumptions would need to be made about (1) the acquisition function, and, perhaps more importantly, (2) the correlation clustering algorithm used. We employ a correlation clustering method based on local search (see Algorithm 5 in the paper), which is guaranteed to efficiently converge to a local minimum of Eq. 1 [5]. However, no guarantees on the quality of the resulting solution are known. Nonetheless, this method performs very well empirically in many practical settings, as mentioned in the paper.
>
> **References**
>
> [1] "Active Learning Literature Survey", Settles 2009.
>
> [2] Sanjoy Dasgupta: Analysis of a greedy active learning strategy. NIPS 2004
>
> [3] Jordan T. Ash, et al. Deep Batch Active Learning by Diverse, Uncertain Gradient Lower Bounds. In International Conference on Learning Representations (ICLR), 2020.
>
> [4] David García-Soriano et al. Query-efficient correlation clustering. In Proceedings of The Web Conference 2020.
>
> [5] Linus Aronsson, Morteza Haghir Chehreghani: Correlation Clustering with Active Learning of Pairwise Similarities. TMLR, 2024.

---

> > ### Comment · Reviewer_gKTJ · 2024-11-19
> >
> > I thank the authors for their detailed responses. In particular, the explanation about the noise level in the different oracles is indeed convincing. I maintain my positive score even if I am not exactly excited about the paper.  Of course, I appreciate the importance of empirical findings and that many widely used algorithms do not provide theoretical guarantees. Yet, it would have been helpful to provide a deeper discussion, maybe even using a wide range of synthetic datasets with carefully analyzed properties and sharing some observations about the achieved approximation. Of course, I realize this is just a conference paper and not all suggestions can be addressed. So, the paper has my lukewarm support.

---

> > > ### Author Response · Authors · 2024-11-20
> > > **Response to Reviewer gKTJ**
> > >
> > > We truly appreciate your feedback on our response and are grateful that you recognize our efforts to address your concerns.
> > >
> > > As you noted, practical information-theoretic acquisition functions typically do not offer theoretical guarantees in many settings, even though they yield state-of-the-art results.
> > >
> > > Regarding the deeper analysis and additional experiments (on synthetic data), in the revised version uploaded after your latest comment, we have added Section C.8 to the Appendix. This section explores how our framework can incorporate features in an innovative way, marking a significant improvement over alternatives. We also demonstrate the impact of feature quality on the effectiveness of active learning. Currently, we have included results from two datasets for this particular study and will soon add further results.

---

### Meta-Review · Area_Chair_7htr · 2024-12-20

**Metareview:**

This paper aims to design correlation clustering methods with a limited number of queries for pairwise similarities between objects.  It introduces active learning and devises four information-theoretic acquisition functions. Specially, it utilizes mean-field approximation to approximate complex quantities and encodes the mean-field approximation using a matrix of assignment probabilities. Experimental results on eight public datasets illustrate the effectiveness. After rebuttal, this manuscript receives divergent scores. Although reviewer gKTJ provides a positive score, he/she acknowledges that his support is lukewarm.


After reviewer discussion, they hold that this manuscript seems to be a practical work and does not balance theoretical insights with practical application. The mean-field approximation scheme seems similar to the commonly used variational inference. Also, the writing can be improved. In view of these, the authors are encouraged to develop more theories to enrich this manuscript in the future.

**Additional Comments On Reviewer Discussion:**

After reviewer discussion, they hold that this manuscript seems to be a practical work and does not balance theoretical insights with practical application. The mean-field approximation scheme seems similar to the commonly used variational inference. Also, the writing can be improved.

---

### Decision · Program_Chairs · 2025-01-22

Reject